# Mitochondrial double-stranded RNA homeostasis depends on cell-cycle progression

Vanessa Xavier[1,2], Silvia Martinelli[1,3], Ryan Corbyn[1], Rachel Pennie[1,3], Kai Rakovic[1,3], Ian R Powley[1,3], Leah Officer-Jones[1,3], Vincenzo Ruscica[4], Alison Galloway[1], Leo M Carlin[1,3], Victoria H Cowling[1,3], John Le Quesne[1,3], Jean-Claude Martinou[2], Thomas MacVicar[1,3]

**Mitochondrial gene expression is a compartmentalised process essential for metabolic function. The replication and transcription of mitochondrial DNA (mtDNA) take place at nucleoids, whereas the subsequent processing and maturation of mitochondrial RNA (mtRNA) and mitoribosome assembly are localised to mitochondrial RNA granules. The bidirectional transcription of circular mtDNA can lead to the hybridisation of polycistronic transcripts and the formation of immunogenic mitochondrial double-stranded RNA (mt-dsRNA). However, the mechanisms that regulate mt-dsRNA localisation and homeostasis are largely unknown. With super-resolution microscopy, we show that mt-dsRNA overlaps with the RNA core and associated proteins of mitochondrial RNA granules but not nucleoids. Mt-dsRNA foci accumulate upon the stimulation of cell proliferation and their abundance depends on mitochondrial ribonucleotide supply by the nucleoside diphosphate kinase, NME6. Consequently, mt-dsRNA foci are profuse in cultured cancer cells and malignant cells of human tumour biopsies. Our results establish a new link between cell proliferation and mitochondrial nucleic acid homeostasis.**

## Introduction

Mitochondria are double membrane organelles that contain multiple copies of their own circular genome. Mitochondrial DNA (mtDNA) is packaged within compact protein–DNA complexes, termed nucleoids, which allows spatial organisation of mtDNA replication and transcription (Garrido et al, 2003; Kukat et al, 2011). Human mtDNA encodes the messenger RNA (mRNA), ribosomal RNAs (rRNA), and transfer RNAs (tRNA) required to synthesise 13 protein subunits of the electron transport chain and ATP synthase necessary for mitochondrial bioenergetic function (Miranda et al, 2022). The mitochondrial genome is transcribed bidirectionally

from both strands of DNA, denoted as heavy (H) and light (L) strands, to produce two almost genome-length polycistronic transcripts, which undergo endonucleolytic processing into mature mRNA, rRNA, and tRNA before further modification (Gustafsson et al, 2016). Newly synthesised mitochondrial RNA (mtRNA) is packaged into spatially defined ribonucleoprotein structures termed mtRNA granules (MRGs), which are often found in close proximity to the nucleoids (Iborra et al, 2004; Antonicka et al, 2013; Jourdain et al, 2013). Numerous proteins involved in mtRNA processing, maturation and mitoribosome assembly colocalise with MRGs (Xavier & Martinou, 2021) and super-resolution microscopy revealed that MRGs are dynamic fluid compartments associated with the mitochondrial inner membrane (Rey et al, 2020).

The bidirectional transcription of H- and L-strand mtDNA generates complementary mtRNA sequences that can hybridise to form mitochondrial double-stranded RNA (mt-dsRNA) (Murphy et al, 1975; Young & Attardi, 1975). The abundance of mt-dsRNA is usually limited by RNA decay, which ensures that L-strand non-coding anti-sense transcripts are kept at very low steady-state levels (Pietras et al, 2018). Single and double-stranded mtRNA are degraded by the mtRNA degradosome complex, which consists of the mitochondrial helicase SUV3 and polynucleotide phosphorylase (PNPase) (Wang et al, 2009; Borowski et al, 2013). Accordingly, suppression of either SUV3 or PNPase drives the accumulation of long sequences of mt-dsRNA (Wang et al, 2009; Dhir et al, 2018).

Analogous to dsRNA of viral origin (Chen & Hur, 2022), long sequences of mt-dsRNA are potent immunogens if exposed to cytosolic effectors of inflammatory responses that sense dsRNA such as MDA5 (Dhir et al, 2018) and PKR (Kim et al, 2018). Suppression of PNPase activity, but curiously not SUV3, elicits a type I interferon response that depends on the release of mt-dsRNA to the cytosol (Dhir et al, 2018). Consequently, mt-dsRNA is implicated in the induction of inflammatory responses in diverse pathophysiological scenarios such as osteoarthritis (Kim et al, 2022), alcohol liver disease (Lee et al, 2020), chronic kidney disease (Zhu et al, 2023), and autoimmune diseases (Yoon et al, 2022; Hooftman et al, 2023).

[1]The CRUK Scotland Institute, Glasgow, UK   [2]Department of Molecular and Cellular Biology, University of Geneva, Genève, Switzerland   [3]School of Cancer Sciences, University of Glasgow, Glasgow, UK   [4]MRC-University of Glasgow Centre for Virus Research, Glasgow, UK

Correspondence: Jean-Claude.Martinou@unige.ch; Thomas.MacVicar@glasgow.ac.uk

Mt-dsRNA foci were detected in cultured cells with normal mtRNA processing and are enhanced in cells that lack functional mtRNA degradosomes (Dhir et al, 2018). Mt-dsRNA also accumulates in the absence of other regulators of mtRNA processing, including the exoribonuclease REXO2 (Szewczyk et al, 2020) and the RNA binding protein GRSF1 (Hensen et al, 2019). Defective poly-adenylation of mitochondrial transcripts also leads to enhanced levels of mt-dsRNA in *Drosophila* (Pajak et al, 2019). Furthermore, mt-dsRNA accumulates in stimulated macrophages upon inhibition of the TCA cycle enzyme fumarate hydratase and ATP synthase (Hooftman et al, 2023), which suggests that mt-dsRNA homeostasis is under metabolic control (Peace et al, 2023). However, despite its relevance to inflammatory disease, the mechanisms that control mt-dsRNA homeostasis remain unclear.

Here, we characterise the sub-mitochondrial distribution of mt-dsRNA foci using super-resolution microscopy and reveal that these discrete structures associate with nascent mtRNA and proteins of MRGs. Furthermore, we show that mt-dsRNA accumulates in transformed cells in a cell proliferation-dependent manner. Our data demonstrate that proliferating cancer cells in culture and in human tumours harbour excessive mt-dsRNA, which may be relevant to mt-dsRNA-dependent innate immune responses.

# Results

### Mt-dsRNA foci associate with MRGs and the degradosome

We investigated whether mt-dsRNA foci are found within mito-chondrial RNA granules (MRGs) or exist in distinct structures. The core of each MRG is composed of nascent single-stranded RNA (ssRNA), which is visualised by incubating cells with bromouridine (BrU) for 1 h before immunodetection with an anti-BrU antibody (Iborra et al, 2004; Jourdain et al, 2013). The BrU label is hidden from the anti-BrU antibody when incorporated in dsRNA and we con-firmed that anti-BrU can only detect BrU-labelled ssRNA (Fig S1A). Direct immunolabelling of dsRNA was performed in the same cells using an antibody (J2) that specifically detects dsRNA of at least 40 nucleotides in length (Schönborn et al, 1991; Weber et al, 2006; Dhir et al, 2018).

We performed super-resolution stimulated emission depletion (STED) microscopy to compare nascent BrU-RNA and dsRNA foci within the mitochondrial network of U2OS cells (Fig 1A). SsRNA and dsRNA foci were observed throughout the mitochondrial network. BrU-RNA and mt-dsRNA foci were similar in size, with average surface areas of 0.017 $\mu m^2$ and 0.019 $\mu m^2$, respectively (Fig 1B). Only ~3% of dsRNA foci co-localised fully with BrU-RNA, whereas ~65% of dsRNA overlapped partially with BrU-RNA and ~32% of dsRNA foci were distinct from any BrU-labelled ssRNA structures (Fig 1C). Several proteins localise to MRGs, including FASTKD2 and GRSF1, which surround BrU-RNA and thus expand the size of MRGs beyond the RNA core (Rey et al, 2020). We found that ~75–80% of BrU-RNA overlapped with FASTKD2/GRSF1 foci, either completely or partially (Fig 1D and E). In contrast, the degree of overlap between dsRNA and FASTKD2/GRSF1 was reduced, and fewer dsRNA foci overlapped with FASTKD2 (~52%) compared with GRSF1 (~65%) (Fig 1E).

We next determined the spatial organisation of the mtRNA degradosome relative to mt-dsRNA foci by STED microscopy. The degradosome proteins, SUV3 and PNPase, localise within mito-chondrial foci, termed D foci, which were identified previously using fluorescently tagged plasmids overexpressing SUV3 and PNPase and conventional confocal microscopy (Borowski et al, 2013; Pietras et al, 2018). We could not detect punctate endogenous PNPase by STED immunofluorescence but specifically detected endogenous SUV3 foci in HeLa and U2OS cells (Fig S1B–D). SUV3 foci were variable in size and most of the small SUV3 foci did not colocalise with ssRNA, dsRNA, or GRSF1 (Fig S1E). However, large SUV3 foci overlapped with dsRNA, ssRNA, and GRSF1 to a similar degree and ~50% of dsRNA foci overlapped partially with SUV3 (Figs 1F and S1D and E). This indicates that only half of MRGs and dsRNA foci are engaged by the degradosome under steady-state conditions at any given time. Little overlap was observed between mtDNA and SUV3 foci of any size, which is consistent with the role of SUV3 in the unwinding of dsRNA and processing of single-stranded mRNA at MRGs (Khidr et al, 2008; Wang et al, 2009; Szczesny et al, 2010; Clemente et al, 2015; van Esveld et al, 2022).

Together, our super-resolution imaging analysis reveals the close association between mt-dsRNA foci and components already described as bona fide components of MRGs. Partial association with SUV3 suggests not all dsRNA foci are immediately unwound and degraded by the SUV3-PNPase degradosome in proliferating cultured cells (Fig 1G).

### Mt-dsRNA accumulates in transformed cells

Interestingly, mt-dsRNA foci are barely detected in slow-dividing and quiescent cell cultures such as primary fibroblasts (Dhir et al, 2018; van Esveld et al, 2022), pancreatic beta cells (Coomans de Brachène et al, 2021) and neurons (Dorrity et al, 2023). We decided to investigate mt-dsRNA homeostasis in untransformed versus transformed cells using an in vitro model of malignant transfor-mation (Hahn et al, 1999). Sequential transduction of CRL-2097 primary human dermal foreskin fibroblasts (WT) with telomerase reverse transcriptase (hTERT), the large T antigen of simian vacuolating virus 40 (LT) oncoprotein and the oncogenic RAS allele, *HRAS*[G12V], produced four cell lines which we termed WT, hTERT, hTERT-LT, and hTERT-LT-RAS (Fig S2A). Sequential transduction resulted in enhanced cell proliferation as expected (Fig S2B). The fully carcinogenic cell line, hTERT-LT-RAS, readily formed colonies (Fig S2C) and exhibited higher oxygen consumption rates (OCR; Fig S2D) and extracellular acidification rates compared with WT fi-broblasts (ECAR; Fig S2D), which reflects up-regulated metabolic activity in highly proliferative fibroblasts. Transformed fibroblasts contained more mitochondrial proteins (Fig S2E), more mtDNA (Fig S2F), and an expanded mitochondrial network (Fig S2G), which together indicate up-regulated mitochondrial biogenesis upon oncogenic transformation of dermal fibroblasts.

WT fibroblasts contained MRGs, as detected with BrU-labelling of ssRNA (Fig 2A) and immunodetection of FASTKD2 and GRSF1 (Fig S3A). BrU-labelled ssRNA foci increased in number after trans-duction with hTERT (Fig 2A and B). In contrast, mt-dsRNA foci were only detected upon additional expression of SV40LT and further increased in number and intensity after the expression of HRAS[G12V]

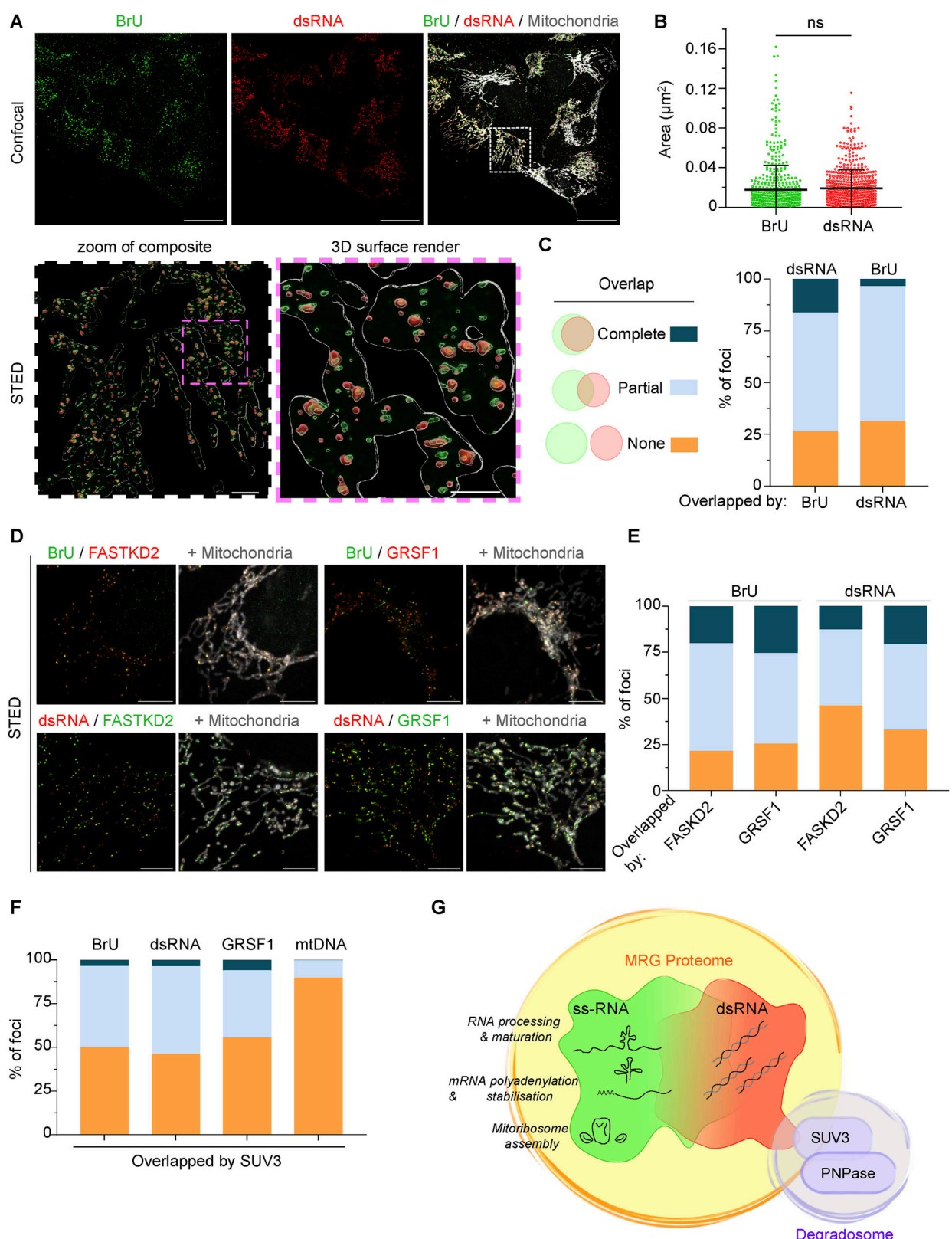

**Figure 1. dsRNA and BrU-labelled mtRNA are closely associated within MRGs.**

**(A)** U2OS cells were treated with BrU for 1 h and immunostained with anti-BrU (green), anti-dsRNA (red), and anti-TOMM20 antibodies (mitochondria; grey) and imaged with confocal microscopy (top row; scale bar: 20 $\mu$m). The indicated region was imaged with stimulated emission depletion (STED) microscopy (bottom row left; scale bar: 2 $\mu$m) and a 3D surface rendering and zoom of the STED image is shown. The white trace defines the boundary of the mitochondrial area (bottom row right; scale bar: 1 $\mu$m).

(Fig 2A and B). Northern blot analysis of the coding and mirror transcripts of *CYTB* and *ND5* revealed increased transcript levels from both strands of mtDNA in transformed cells (Fig 2C), suggesting either increased transcription and/or decreased degradation of transcripts in the transformed cells. Importantly, whereas mt-dsRNA foci were normally undetectable in WT fibroblasts, silencing of the degradosome enzymes SUV3 or PNPase by siRNA resulted in their accumulation (Figs 2D and S3B). This is analogous to the accumulation of mt-dsRNA in fibroblasts from patients with pathological mutations in *PNPT1*; the gene that encodes PNPase (Dhir et al, 2018). Our data demonstrate that mt-dsRNA species are generated in untransformed cells but sufficiently degraded to undetectable levels. The hTERT-LT and hTERT-LT-RAS cell lines accumulated mt-dsRNA (Fig 2A and B) despite containing higher levels of PNPase and SUV3 compared with WT cells (Fig S2E), which argues that transformed fibroblasts produce more dsRNA compared with WT cells.

We and others recently demonstrated that the mitochondrial nucleotide diphosphate kinase, NME6, supplies ribonucleoside triphosphates (rNTPs) to sustain mitochondrial transcript levels (Grotehans et al, 2023; Kramer et al, 2023). NME6 associates with MRGs via interaction with RCC1L within the RNA pseudouridylation module (Antonicka et al, 2020) and shows partial colocalisation with Bru-labelled ssRNA (Grotehans et al, 2023). We noticed that NME6 levels increased dramatically in hTERT-LT and hTERT-LT-RAS cells (Fig S2E), in line with increased mitochondrial biogenesis and demand for rNTPs. Strikingly, mt-dsRNA was strongly depleted in *NME6* knockout HeLa cells and restored upon expression of NME6-MycFlag but not kinase-dead NME6[H137N]-MycFlag (Fig 2E and F). Transient depletion of NME6 by siRNA was also sufficient to block the accumulation of mt-dsRNA in HeLa and U2OS cells (Fig S3C). Exogenous supplementation with nucleosides boosted rNTP levels in cells lacking NME6 (Grotehans et al, 2023; Kramer et al, 2023) and led to the partial restoration of mt-dsRNA foci in *NME6* knockout cells (Fig 2G and H). However, nucleoside supplementation in WT fibroblasts did not increase mt-dsRNA levels, which indicates that boosting mitochondrial rNTPs alone is insufficient to drive mt-dsRNA foci formation (Fig S3D). Collectively, these data reveal that mt-dsRNA accumulates in rapidly dividing malignant cells, which is supported by the supply of mitochondrial rNTPs by NME6 for RNA synthesis.

## The accumulation of mt-dsRNA depends on cell-cycle progression

We hypothesised that mt-dsRNA homeostasis is linked to cell proliferation, which is increased in cancer cells. To test this, we halted HeLa cell-cycle progression by double thymidine block (DTB) treatment and monitored mt-dsRNA levels (Fig 3A and B). DTB treatment reduced mt-dsRNA levels in WT cells to a level comparable with *NME6* knockout cells. Conversely, mt-dsRNA levels were unaffected by DTB treatment in *NME6* knockout cells. Release from the DTB by media exchange permitted cells to re-enter the cell cycle in S phase and led to the synthesis of mt-dsRNA in WT cells within 4 h but had minimal effect on mt-dsRNA in *NME6* KO cells (Fig 3A and B). We next determined the cell-cycle status of unsynchronised HeLa cells by EdU and cyclin-A staining and found higher levels of mt-dsRNA in S and G2 phase cells compared with G1 (Fig 3C and D). *NME6*-depleted HeLa cells in S and G2 phases also contained slightly more mt-dsRNA compared with G1 but still failed to reach the levels of WT cells (Fig 3C and D). It is important to note that the surface area of cells in S and G2 phase is greater on average compared with cells in G1 (Fig 3E), which correlates with total levels of mt-dsRNA (Fig 3D). Together, these data demonstrate that mt-dsRNA levels vary according to the cell-cycle phase in cultured cells and arrest of the cell-cycle depletes mt-dsRNA.

We next tested whether stimulation of cell proliferation can raise mt-dsRNA levels. Growth factors induce fibroblast proliferation (Yun et al, 2010) and have been shown to induce mitochondrial biogenesis in various cell types (Echave et al, 2009; Zhang et al, 2022). We stimulated the proliferation of hTERT fibroblasts with FGF, which we could block by co-treatment with a specific inhibitor of nuclear DNA polymerases called aphidicolin (Sasaki et al, 2017) (Fig 4A). Because the inhibition of nuclear DNA replication leads to apoptosis in these cells, we also added z-VAD, a pan-caspase inhibitor, to the medium of aphidicolin-treated cells to prevent cell death. FGF-stimulated cell proliferation was accompanied by an increase in nascent mtRNA (Fig S4A) and mt-dsRNA foci (Fig 4B) in hTERT fibroblasts, which was completely abolished in cell cycle-arrested cells treated with aphidicolin (Fig 4C and D). Consistent with the immunodetection of dsRNA, strand-specific RT-qPCR (Kim et al, 2023) revealed an accumulation of heavy and light strand transcripts in FGF-treated fibroblasts, which was reversed by co-treatment with aphidicolin (Fig 4E). Similarly, Northern blot analysis of the coding and mirror transcripts of *CYTB* revealed increased transcript levels from both strands of mtDNA in FGF-treated fibroblasts (Fig S4B). The increase in mtRNA and mt-dsRNA upon FGF treatment was independent of mtDNA levels (Fig 4F) and mitochondrial area (Fig S4C), both of which were increased further by aphidicolin. The increase in mtDNA likely reflects enhanced mtDNA replication coupled with the larger cell volumes of aphidicolin-treated cells (Seel et al, 2023). Levels of the mitochondrial RNA polymerase (POLRMT), NME6, PNPase, and SUV3 were unchanged

---

**(B)** Beeswarm plot of individual BrU and dsRNA foci area in U2OS cells imaged by STED microscopy. Horizontal lines indicate the mean value and error bars indicate the SD. Number of foci measured: BrU = 537; dsRNA = 575; number of cells = 7 from one culture. Welch's unpaired *t* test; ns, not significant. **(B, C)** The degree of overlap between BrU and dsRNA foci represented in (B) was categorised as shown and calculated as a percentage of the total number of BrU and dsRNA foci measured. **(D)** U2OS cells transfected with mito-EYFP (grey) were immunostained with the indicated antibodies and imaged by STED microscopy (scale bar: 20 $\mu$m). **(C, E)** The percentage of BrU and dsRNA foci overlapped by FASTKD2 and GRSF1 foci categorised as in (C). BrU versus FASTKD2: number of foci measured = 315; number of cells = 10; BrU versus GRSF1: number of foci measured = 275; number of cells = 5; dsRNA versus FASTKD2: number of foci measured = 1,806; number of cells = 10; dsRNA versus GRSF1: number of foci measured = 1,432; number of cells = 8. **(C, F)** Percentage of by BrU, dsRNA, GRSF1, or mtDNA foci overlapped by SUV3 foci categorised as in (C). BrU versus SUV3: number of foci measured = 958; number of cells = 4; dsRNA versus SUV3: number of foci measured = 1,189; number of cells = 4; GRSF1 versus SUV3: number of foci measured = 868; number of cells = 4; mtDNA versus SUV3: number of foci measured = 773; number of cells = 4. **(G)** Model depicting the sub-compartmentalisation of single-stranded RNA and dsRNA in the MRG with associated RNA processing functions. Approximately half of MRGs and dsRNA foci associate with SUV3, which interacts with PNPase within the degradosome.

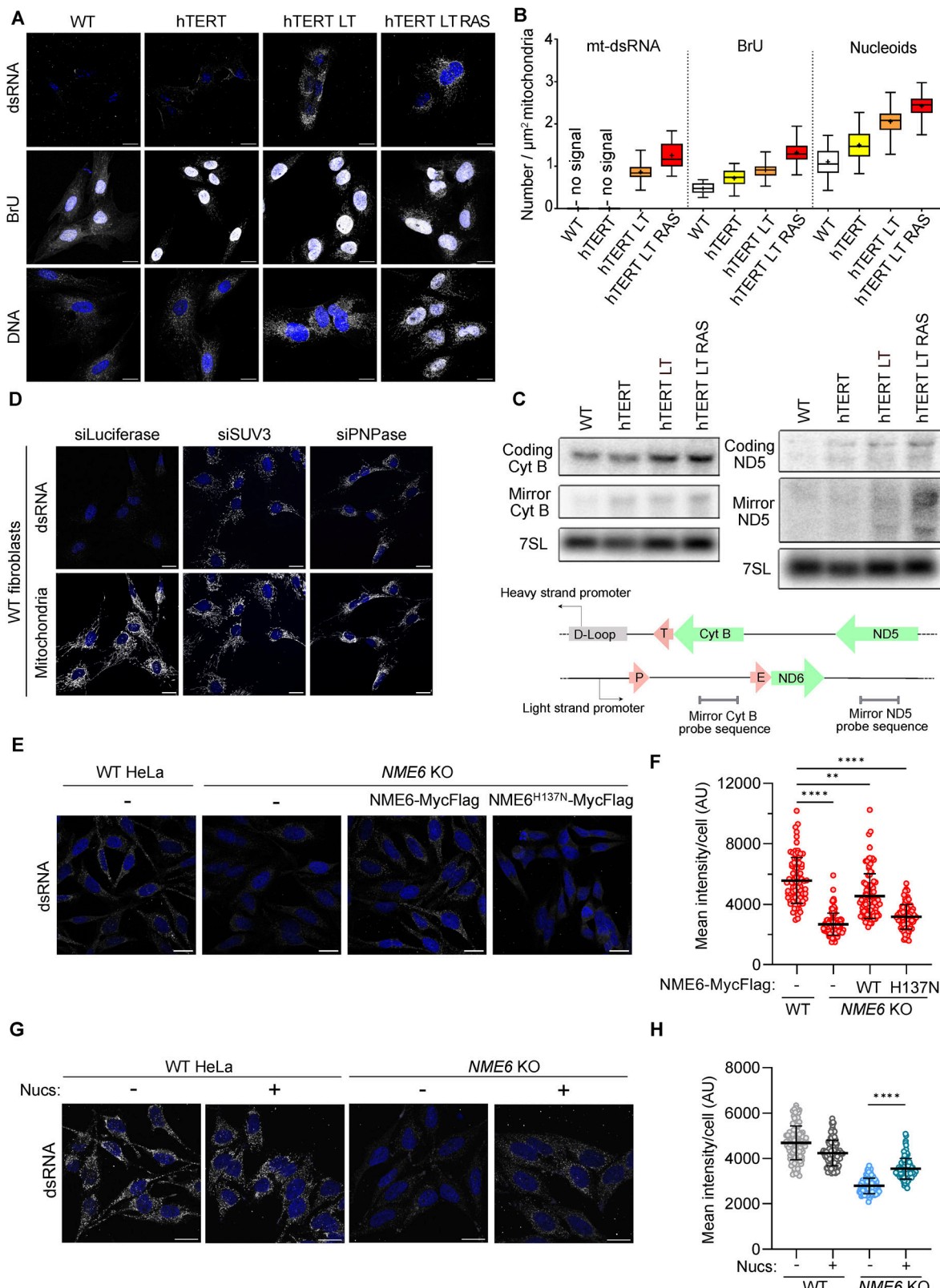

**Figure 2. Mitochondrial dsRNA foci accumulate upon oncogenic transformation.**
**(A)** Immunofluorescence of dsRNA, BrU, and DNA in the indicated fibroblast cell lines treated with BrU for 1 h and imaged by confocal microscopy. DAPI staining shown in blue (scale bar: 20 μm). **(B)** Number of mitochondrial dsRNA foci, BrU foci, and nucleoids per μm² of mitochondria determined by immunofluorescence and confocal microscopy. The mitochondrial network was imaged using anti-TOMM20 antibody. Box and whiskers plot represent the number of foci per μm² mitochondria for each cell.

upon FGF treatment (Fig 4G), which indicates that acute up-regulation of mitochondrial RNA synthesis and mt-dsRNA is not a consequence of altered expression of mitochondrial transcription machinery or degradosome components. Together, our data show that mt-dsRNA foci accumulate upon stimulation of cell prolifer-ation and that acute inhibition of nuclear DNA replication hinders the maintenance of mtRNA but not mtDNA.

### Visualization of mt-dsRNA in human tumours

The strong link between mt-dsRNA homeostasis, cell transforma-tion and proliferation encouraged us to examine the levels of dsRNA in human tumours. The anti-dsRNA monoclonal antibody clone 9D5 was validated recently in formalin-fixed paraffin-embedded (FFPE) tissue where it outperforms the J2 antibody in detecting viral dsRNA in clinical samples (Thomsen et al, 2023). The 9D5 antibody detected mt-dsRNA foci in HeLa cells as shown by colocalisation with J2 (Fig 5A). 9D5 and J2 staining of mt-dsRNA was absent after the inhibition of mitochondrial transcription by in-hibitor of mitochondrial transcription 1 (IMT1) (Bonekamp et al, 2020) or treatment of the dsRNA-specific ribonuclease RNAse III (Figs 5B and S5A and B). We therefore used the 9D5 antibody to detect dsRNA in normal and tumour tissue sections from patients with colorectal and lung adenocarcinomas by immunohisto-chemistry (IHC). Remarkably, intracellular dsRNA staining was most intense in the dysplastic epithelial cells in the colon (Fig 5C) and lung adenocarcinoma cells (Fig 5D). The intensity of dsRNA cor-related positively with the mitochondrial protein ATP5A, whereas normal lung and colon epithelium were devoid of detectable dsRNA. In both cases, pre-treatment of FFPE sections with RNAse III strongly reduced the signal intensity, which further confirmed the specificity of 9D5 for dsRNA (Fig 5C and D).

To corroborate our findings, we performed fluorescent multiplex staining of dsRNA (9D5), ATP5A, DAPI, and pan-cytokeratin (CK) on lung adenocarcinoma tissue microarrays that accommodated tis-sue cores from 80 patients. We developed a supervised machine learning algorithm from pathologist annotations of tumour, stroma, and necrosis regions to perform tissue segmentation (Fig 6A). In agreement with IHC staining (Fig 5D), dsRNA was predominantly found in tumour epithelial cells, where we observed a high degree of overlap with mitochondrial ATP5A (Fig 6B and C). The median cellular cytoplasmic dsRNA intensity was significantly higher in tumour regions compared with stromal regions across the tissue cores (Fig 6D). Nevertheless, heterogenous dsRNA staining was

detected within the stroma of many lung tumours, as shown by the broad distribution of dsRNA intensity across stromal cells (Fig 6E). Indeed, we observed high levels of dsRNA and ATP5A in stromal cells that appeared to be predominantly macrophages and other immune cells (Fig S6A), indicating that some stromal cell pop-ulations also harbour detectable mt-dsRNA. The epithelia of normal lung tissue were devoid of dsRNA as expected (Fig S6B). Collectively, these data couple mt-dsRNA accumulation to malignancy in human lung adenocarcinoma.

## Discussion

MRGs have been described as compartments in which mitochon-drial transcripts accumulate to be processed and matured (Iborra et al, 2004; Antonicka et al, 2013; Jourdain et al, 2013). These as-sumptions are based on RNA and protein composition of these structures (Xavier & Martinou, 2021). In addition, three protein modules that participate in pseudouridylation, large ribosomal subunit assembly and RNA processing are also found within MRGs, suggesting that they are also the place for ribosome assembly (Antonicka et al, 2017; Zaganelli et al, 2017). We report the existence of a new MRG domain that contains dsRNA. Super-resolution mi-croscopy allowed us to define dsRNA foci in contact with bona fide components of MRGs, including ssRNA, FASTKD2 and GRSF1. Ap-proximately half of mt-dsRNA foci associate with SUV3, which likely indicate these dsRNA are soon to be degraded. The presence of dsRNA in ribonucleoprotein compartments is not unique to mi-tochondria because dsRNA can also be detected in cytosolic stress-induced RNA-protein condensates, which help sequester endog-enous immunogenic dsRNA (Maharana et al, 2022). The cohabita-tion of ssRNA and dsRNA is not a universal feature of cytosolic RNA granules, which often exclude viral dsRNA (Langereis et al, 2013; Oh et al, 2016; Paget et al, 2023).

Our data reveal that the steady-state level of mt-dsRNA corre-lates positively with cell proliferation, and mt-dsRNA is diminished upon cell-cycle arrest. This likely explains the undetectable levels of mt-dsRNA in slow-dividing primary fibroblasts and quiescent cells such as pancreatic beta cells (Coomans de Brachène et al, 2021). Cancer cells by comparison accumulate dsRNA foci throughout their mitochondrial network. Malignant transformation or FGF treatment triggered the build-up of mt-dsRNA in dermal fibroblasts, which may result from enhanced mtRNA synthesis. FGF treatment enhanced the levels of mtRNA but not mtDNA, suggesting

Whiskers represent minimum and maximum values. Boxes extend from the 25th to the 75th percentile with the median plotted in the middle. "+" indicates the mean value (N = 30 cells from two independent cultures). **(C)** Northern blot of mRNA transcripts of the coding and mirror regions of *CYTB* and *ND5* in the fibroblast cell lines (top). Nuclear 7SL RNA was used as a loading control. Schematic representing the coding and mirror regions that were probed for *CYTB* and *ND5* (bottom). Coding genes: green arrows. Coding regions for tRNAs: pink arrows. **(D)** Immunofluorescence of dsRNA in WT fibroblasts treated with the indicated siRNA for 48 h and imaged by confocal microscopy. The mitochondrial network was immunostained with anti-TOMM20. DAPI staining shown in blue (scale bar: 20 μm). **(E)** Immunofluorescence of dsRNA in WT, NME6 KO, and NME6 KO HeLa cells expressing NME6-MycFlag or NME6^H137N^-MycFlag and imaged by confocal microscopy. DAPI staining shown in blue (scale bar: 20 μm). **(E, F)** Scatter plot of mean mt-dsRNA intensity per cell quantified from confocal images as shown in (E) (N = 100–120 cells from two independent cultures). The one-way ANOVA test was used to determine *P*-values compared with WT. *P*-value (WT versus *NME6* KO) < 0.0001, *P*-value (WT versus *NME6* KO + WT) = 0.0013, *P*-value (WT versus *NME6* KO + H137N) < 0.0001. Horizontal lines indicate the mean value and error bars indicate the SD. **(G)** Immunofluorescence of dsRNA in WT and NME6 KO HeLa cells incubated with 100 μM nucleosides for 5 d and imaged by confocal microscopy. DAPI staining shown in blue (scale bar: 20 μm). **(G, H)** Scatter plot of mean mt-dsRNA intensity per cell quantified from confocal images as shown in (G) (N = 100–120 cells from two independent cultures). The Mann-Whitney *t* test was used to determine the *P*-value between *NME6* KO versus *NME6* KO + nuc. *P*-value < 0.0001. Horizontal lines indicate the mean value and error bars indicate the SD. Source data are available for this figure.

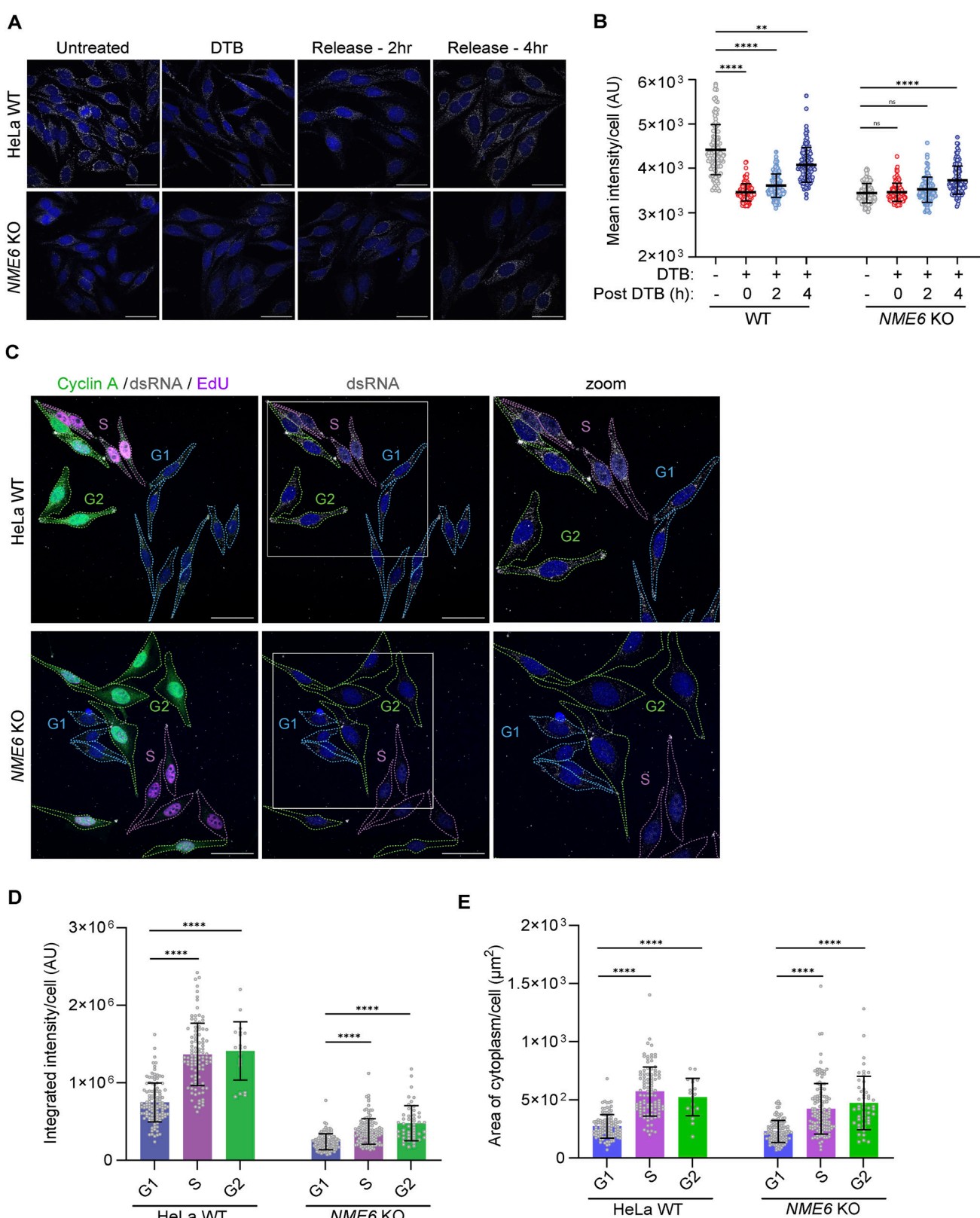

**Figure 3.  Mt-dsRNA homeostasis is dependent on cell-cycle progression.**
**(A)** Immunofluorescence of dsRNA in WT HeLa and *NME6* KO cells after double thymidine block (DTB) treatment and subsequent release for 2 or 4 h imaged by confocal microscopy. DAPI staining is shown in blue (scale bar: 20 μm). **(A, B)** Scatter plot of mean dsRNA intensity per cell quantified from confocal images as shown in (A) (N = 100–110 cells from two independent cultures). A one-way ANOVA test was used to determine *P*-values compared with untreated samples in each cell line. *P*-value (WT

that FGF-treated fibroblasts contain a greater proportion of nucleoids engaged in transcription (Bruser et al, 2021; Tan et al, 2023). Whereas progress through the cell cycle is required for mt-dsRNA synthesis, it is unclear to what degree mt-dsRNA homeostasis is coordinated with each stage of the cell cycle. Mitochondrial transcription rate and transcript levels were reported to be highest in G1 and G2 of the HeLa cell cycle (Chatre & Ricchetti, 2013), but this does not appear to be the case for mt-dsRNA. We found mitochondrial transcription and dsRNA accumulation to be independent of mtDNA homeostasis but to be dependent on nuclear DNA replication. This highlights a link between DNA replication in the nucleus and transcription in mitochondria. Finally, cellular metabolic status and TCA cycle activity have been shown to impact mt-dsRNA homeostasis (Hooftman et al, 2023), and therefore, the levels of mt-dsRNA levels could be linked to the metabolic status of proliferating cells, including the supply of mitochondrial ribonucleotides.

Mt-dsRNA is diminished in proliferating cells lacking NME6, demonstrating a dependency on NTP supply and transcription for the formation of mt-dsRNA foci. However, the level of nucleotides present in NME6-depleted mitochondria is still sufficient to allow at least the initiation of transcription (Grotehans et al, 2023; Kramer et al, 2023). We expect that diminished mitochondrial NTPs in the absence of NME6 prevent complete polycistronic transcripts to be synthesised (Grotehans et al, 2023), which would result in fewer hybridisation events between longer strands of ssRNA to generate dsRNA. In addition, the regulation of mRNA stability by NME6 may also affect dsRNA homeostasis (Kramer et al, 2023).

The clear presence of endogenous dsRNA in the malignant cells of human tumours brings the significance and relevance of mt-dsRNA homeostasis into focus. Efficient degradation of mt-dsRNA is required to minimise its exposure to cytosolic dsRNA sensors, which can otherwise drive chronic and pathogenic inflammation (Dhir et al, 2018; Lee et al, 2020; Kim et al, 2022; Zhu et al, 2023). It remains to be seen whether mt-dsRNA contributes to inflammation during tumour development. We also speculate that the build-up of immunostimulatory mt-dsRNA in proliferating cancer cells could be leveraged for therapeutic advantage. For instance, chemotherapy and radiotherapy regimens can trigger endogenous dsRNA-mediated inflammatory responses, which synergise with immunotherapy (Chiappinelli et al, 2015; Roulois et al, 2015; Ranoa et al, 2016). Interestingly, mtRNA is released into the cytosol of cancer cells in culture upon ionizing radiation (Tigano et al, 2021) and mt-dsRNA may work alongside other non-mitochondrial sources of endogenous dsRNA to drive inflammatory responses to epigenetic and genotoxic stress (Chen & Hur, 2022). NME6 was recently implicated as a positive regulator of inflammatory signalling (Ernst

et al, 2021) and future studies will determine if this is linked to mt-dsRNA homeostasis.

Our study has characterised the spatial regulation of dsRNA foci within mitochondria and illuminated a link between mt-dsRNA homeostasis and cellular proliferation. The build-up of mt-dsRNA appears to be a novel marker of cell malignancy and it will be interesting to determine whether proliferating cells accumulate mt-dsRNA in healthy tissue and normal physiological contexts.

# Materials and Methods

### Cell culture and treatments

Primary and transformed human dermal foreskin fibroblasts, U20S and HeLa cell lines were cultured in DMEM (11995073; Thermo Fisher Scientific) supplemented with 10% heat-inactivated FBS (A5256801; Thermo Fisher Scientific) and GlutaMAX (35050061; Thermo Fisher Scientific) in 5% $CO_2$ at 37°C. All reagents were purchased from Invitrogen Life Technologies. For staining of nascent mtRNA, 1-h pulse of 5 mM of 5'-bromouridine (BrU) (850187; Sigma-Aldrich) resuspended in PBS was added to cells before fixation. For RNA interference, 20 nM of siRNA was reverse transfected in cells using Lipofectamine RNAiMax (Life Technologies) for 48 h, according to the manufacturer's instructions. For FGF and aphidicolin treatments, 10 ng/ml of recombinant human FGF-2 (100-18B; Prepotech) was added with or without the addition of 6 $\mu$M aphidicolin (HY-N6733; MedChemExpress) and 50 $\mu$M of pan-caspase inhibitor Z-VAD (HY-16658; MedChemExpress). For IMT1 treatment, 10 $\mu$M of IMT1 (HY-134539; MedChemExpress) was added to cells for 3 h before fixation. For nucleoside treatments, 100 $\mu$M of nucleoside mix (ES-008-D; EmbryoMax Nucleosides, Millipore) was added to cells for 5 d before fixation. All cell lines used in the study were regularly tested for mycoplasma contamination.

### Transformation of primary dermal fibroblasts

Human primary foreskin dermal fibroblast cells, CRL-2097 (ATCC) were transduced with retrovirus particles generated with plasmids encoding pBABE-hTERT-puromycin, pBABE-SV40LT-neomycin, and pMSCV-HRasGV12-blasticidine. Retroviruses were produced in HEK293 cells by co-transfection of each construct with the packaging plasmids. After 48 h, the culture medium containing the viral particles was filtered and added directly to the fibroblast cultures. After infection for 48 h, 1 $\mu$g/ml of the appropriate selection reagent was added, and cells were cultured for a further 7 d. In each case, a

---

versus WT-DTB) < 0.0001, *P*-value (WT versus WT −2 h) < 0.0001, *P*-value (WT versus WT-4 h) = 0.008, *P*-value (KO versus KO-DTB) > 0.9999, *P*-value (KO versus KO-2 h) = 0.1011, *P*-value (KO versus KO-4 h) < 0.0001. Horizontal lines indicate the mean value and error bars indicate the SD. **(C)** Unsynchronised cell populations of WT HeLa and *NME6* KO imaged by confocal microscopy. Cells in G1 (blue), S (magenta), and G2 (green) phases are delineated according to DAPI, EdU, and cyclin-A detection, respectively (scale bar: 20 $\mu$m). **(C, D)** Scatter plot of mt-dsRNA integrated intensity per cell quantified from confocal images as shown in (C) (N = 100–110 cells from two independent cultures). A one-way ANOVA test was used to determine *P*-values compared with WT and KO for cell-cycle phases in each cell line. *P*-value (WT G1 versus WT S) < 0.0001, *P*-value (WT G1 versus WT G2) < 0.0001, *P*-value (KO G1 versus KO S) < 0.0001, *P*-value (KO G1 versus KO G2) <0.0001. Bars indicate the mean and the error bars indicate the SD. **(C, E)** Scatter plot of the cytoplasmic area per cell quantified from confocal images as shown in (C) (N = 100–110 cells from two independent cultures). A one-way ANOVA test was used to determine *P*-values compared with WT and KO for cell-cycle phases in each cell line. *P*-value (WT G1 versus WT S) < 0.0001, *P*-value (WT G1 versus WT G2) < 0.0001, *P*-value (KO G1 versus KO S) < 0.0001, *P*-value (KO G1 versus KO G2) <0.0001. Bars indicate the mean and the error bars indicate the SD.

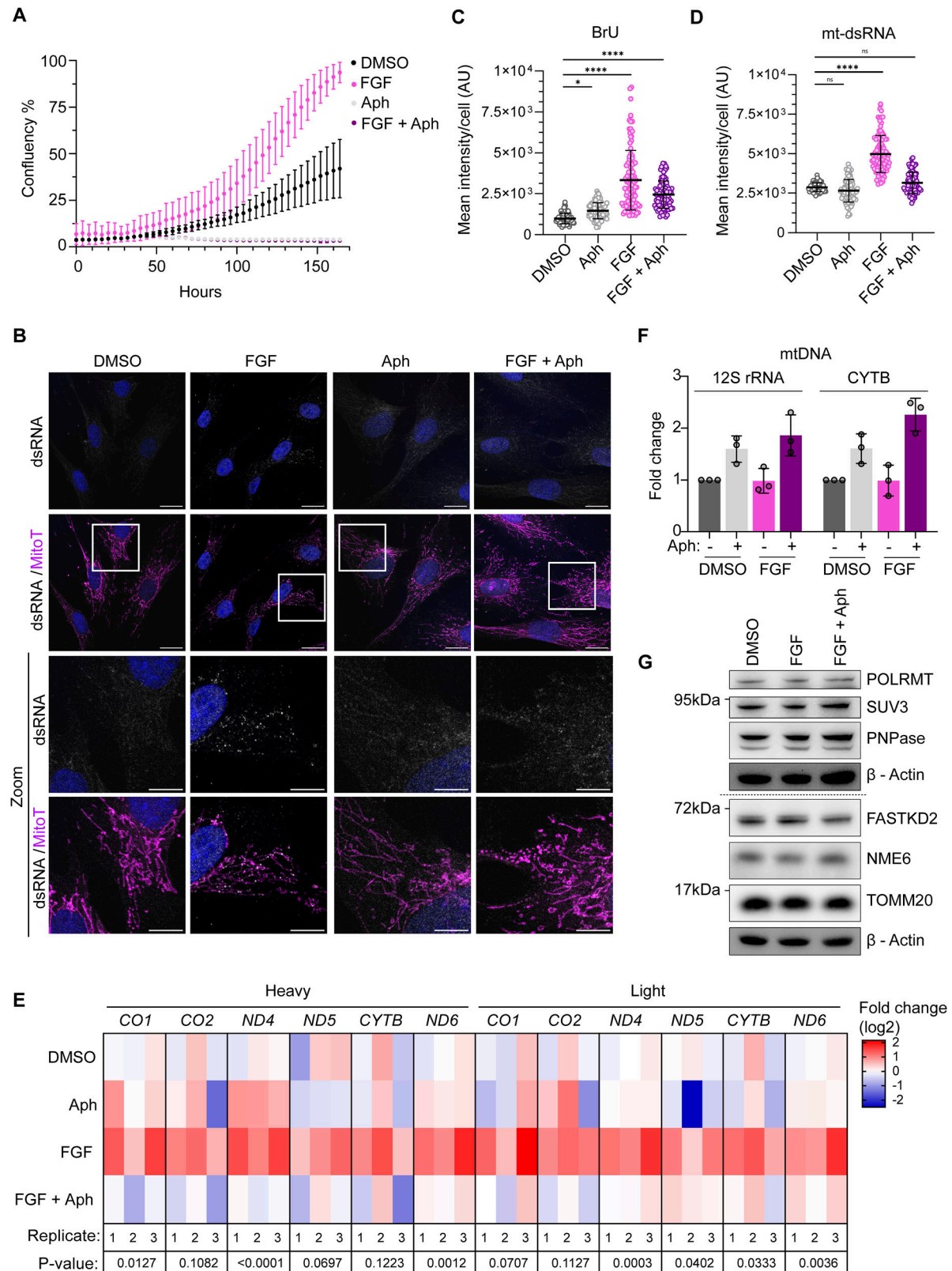

**Figure 4. Synthesis of mitochondrial single-stranded RNA and dsRNA is coupled to cellular proliferation in fibroblasts.**
**(A)** hTERT fibroblasts were treated with basic FGF (10 ng/ml), aphidicolin (Aph; 6 μM), or a combination of both. Cell proliferation was measured by confluency (N = 3 independent cultures; points indicate mean and error bars represent SD). **(B)** Immunofluorescence of dsRNA in hTERT fibroblasts treated with FGF/Aph for 48 h and BrU for 1 h and imaged by confocal microscopy. The mitochondrial network was labelled with MitoTracker DeepRed (MitoT; magenta). DAPI staining shown in blue (scale bar:

clonal population of cells was established before subsequent transduction.

## Immunofluorescence

All cells were seeded at a density of $3 \times 10^5$ cells per well in a six-well culture plate and left to attach for 24 h. Where indicated, cells were incubated with Mitotracker DeepRed (M22426; Life Technologies) for 15 min at a concentration of 1:2,000 before fixation. Cells were fixed for 15 min in 4% PFA that had been pre-warmed to 37°C, followed by three washes in PBS. All subsequent steps were performed in immunofluorescence buffer (IF buffer) consisting of 5% pre-immune goat serum (PCN5000; Thermo Fisher Scientific), 0.15% Triton X-100 (X100; Sigma-Aldrich) in PBS at RT. Permeabilisation and blocking were performed in a single step by incubating the fixed cells for 30 min in IF buffer. Primary antibodies at the appropriate dilution (Table 1) were added to the fixed cells for 1.5 h. Cells were washed three times in PBS and incubated with secondary antibodies (Table 1) for 45 min. Nuclei were stained by incubation with DAPI for 5 min. FluorSave reagent (345789; Millipore) was used to fix coverslips of cells onto slides. For immunofluorescence of nuclease treated cells, U2OS cells were fixed as above and permeabilised in 0.25% Triton X-100 (X100; Sigma-Aldrich) in PBS for 30 min. 40 U/ml of ShortCut RNAse III (M0245S; New England BioLabs) diluted in PBS containing 5 mM MgCl$_2$ was added onto the samples for 30 min at 37°C, followed by three washes in PBS. The samples were then incubated in IF buffer and antibody staining performed as above.

## Image acquisition

For confocal imaging, the Zeiss LSM710 upright confocal microscope equipped with a Plan-Apochromat oil objective (63X, NA 1.40) was used. The pinhole was opened to 1 AU for image acquisition. For STED imaging, the Leica TCS SP8 STED 3X inverted microscope equipped with an HC Plan-Apochromat glycerol motC STED W objective (93X, NA 1.30) was used. The microscope is equipped with a white laser (470–670 nm) and 592 nm and 775 nm depletion lasers for STED. Both STED depletion lasers were set to 55% of maximum power. The pinhole was opened to 1 AU for image acquisition. Lightning mode (Leica) was used to deconvolve STED images. For super-resolution imaging, the Zeiss LSM 880 Airyscan Confocal microscope equipped with the Plan-Apochromat 40×/1.3 Oil DIC M27 objective was used with the following three lasers: 633 nm (BP 570–620 + LP 645), 561 nm (BP 420–480 + BP 495–620), and 488 nm (BP 420–480 + BP 495–550).

## Image analysis

STED images were analysed using a bespoke analysis pipeline. Before analysis, the images were processed using Fiji (Version 1.54). Making use of the selection tool, a binary mask was made of the mitochondrial matrix. Manual thresholding was used to define regions of interest (ROIs) for the imaging channels corresponding to the protein clouds and the RNA granules. The binary mask and ROIs were exported and analysed using a bespoke Python script developed using a jupyter notebook (Kluyver et al, 2016). The script first filtered the ROIs for both the protein and RNA granule channels to remove those that did not lie within the binary mask created for the mitochondrial matrix. After this, the remaining ROIs were analysed to determine their physical characteristics, such as area, major and minor axis length, and eccentricity. In addition, the ROIs were analysed to determine if there was overlap between the protein clouds and the RNA granules, with the area of overlap recorded as a percentage of the overall ROI area.

Area of mitochondrial network was quantified using the MiNA toolset (Valente et al, 2017) on Fiji software (Version 1.54). Immunostaining of the mitochondrial network was binarized and used to generate mitochondrial masks to measure the area of the mitochondrial network per cell in Fig S2G and the number of BrU and mt-dsRNA foci for the fibroblast cell line in Fig 2B.

Intensity of mt-dsRNA in Figs 2E and G and 3B–E was measured using Fiji software. Staining of the mitochondrial network was minimally processed using "unsharp Mask" set to radius = 100 and mask = 0.7. The resulting mitochondrial mask was used to create ROIs to measure the mean intensity mt-dsRNA within the mitochondrial network per cell. Imaris software (Version 10.1) was used to create a mitochondrial mask for each image based on automatic thresholding. The mitochondrial mask was used to create a mitochondrial surface which was then manually grouped by cells. The mitochondrial network per cell was then assigned as individual ROIs. These ROIs were then used to measure the area of mitochondrial network per cell in Fig S4C and the mean intensities of mt-dsRNA and BrU per cell in Fig 4C and D, S3B, and S5A and B.

## Western blotting

Cells were pelleted and resuspended in RIPA buffer (150 mM NaCl, 1% Triton X-100, 0.5% sodium deoxycholate, 0.1% SDS, 50 mM Tris, pH 8.0) supplemented with protease inhibitors (Roche, CO-RO) according to the manufacturer's instructions for 30 min on ice. The lysate was cleared by centrifugation at 16,000 $g$ for 15 min at

20 $\mu$m). **(C)** Scatter plot of mean BrU intensity per cell quantified from confocal images as shown in (Fig S4A) (N = 92–111 cells from two independent cultures). A one-way ANOVA test was used to determine $P$-values compared with DMSO. $P$-value (DMSO versus Aph) = 0.0130, $P$-value (DMSO versus FGF) < 0.0001, $P$-value (DMSO versus FGF + Aph) < 0.0001. Horizontal lines indicate the mean value and error bars indicate the SD. **(D)** Scatter plot of mean mt-dsRNA intensity per cell quantified from confocal images as shown in **(B)** (N = 89–111 cells from two independent cultures). A one-way ANOVA test was used to determine P-values compared to DMSO. $P$-value (DMSO versus Aph) = 0.2653, $P$-value (DMSO versus FGF) < 0.0001, $P$-value (DMSO versus FGF + Aph) = 0.0735. Horizontal lines indicate the mean value and error bars indicate the SD. **(E)** Heatmap showing relative RNA expression of heavy and light mitochondrial transcripts from hTERT fibroblasts treated with FGF/Aph for 48 h as measured by strand-specific RT-qPCR. Fold changes versus mean DMSO values for each transcript are shown. C$_q$ values were normalised against *GAPDH* (N = 3 independent cultures). One-way ANOVA analysis was performed for each transcript among the four conditions and the resulting $P$-value is indicated below. **(F)** mtDNA levels of hTERT fibroblasts treated with FGF/Aph for 48 h as measured by qPCR of the indicated mtDNA regions, *MT-RNR1* and *MT-CYTB*. C$_q$ values were normalised against nuclear *ACTB* (N = 3 independent cultures). Bars indicate the mean and the error bars represent the SD. Individual data points are shown as grey circles. **(G)** Immunoblot analysis of mitochondrial and MRG-associated proteins in whole-cell lysates obtained from hTERT fibroblasts treated with FGF/Aph for 48 h. Samples were blotted on different membranes which are indicated by the dotted lines.
Source data are available for this figure.

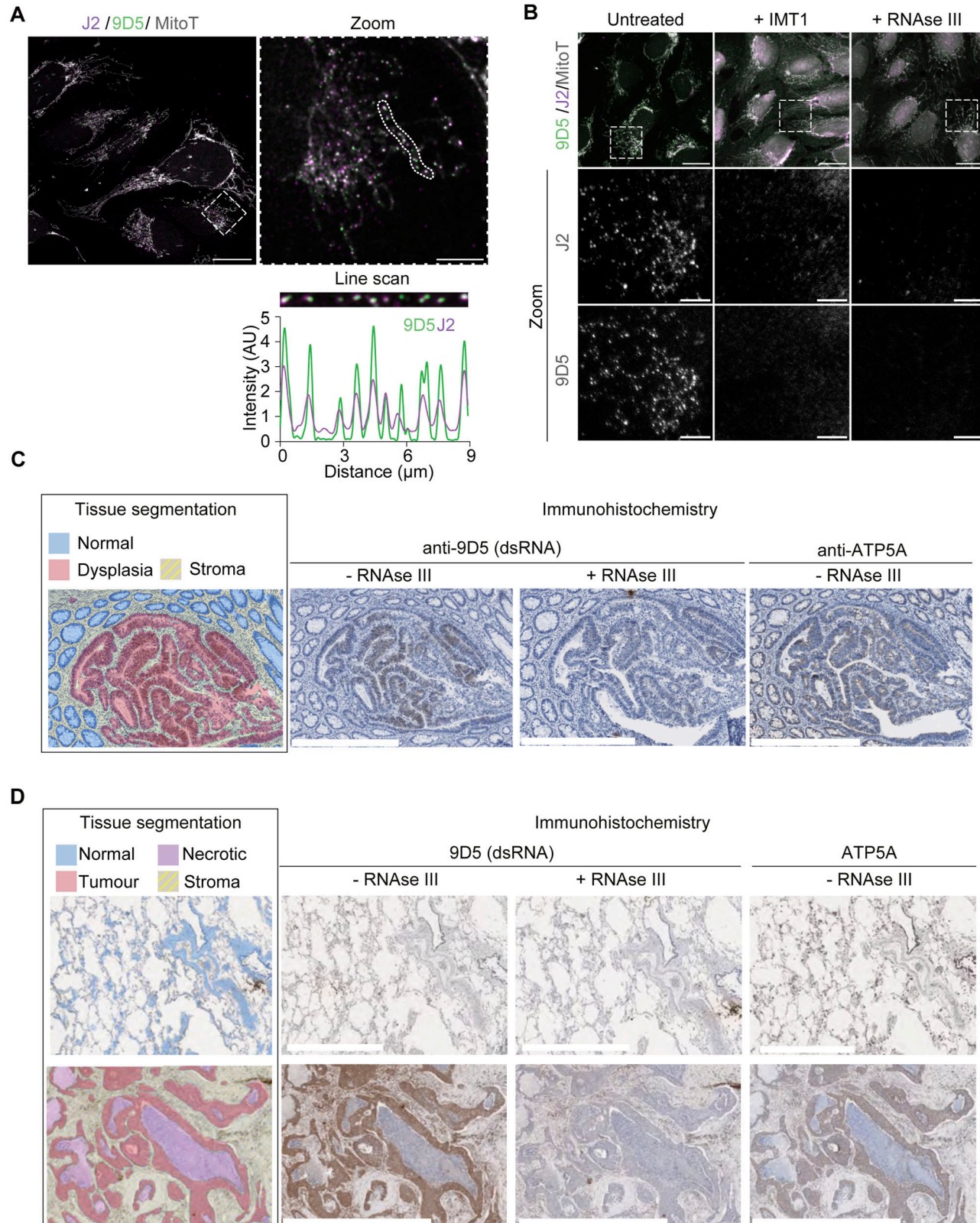

**Figure 5. Immunodetection of dsRNA in human tissue.**
**(A)** Immunofluorescence of mt-dsRNA with anti-J2 and anti-9D5 antibodies in U2OS cells imaged by confocal microscopy. The mitochondrial network was labelled with MitoTracker (MitoT) DeepRed (scale bar: 20 μm). The region indicated by a white dotted box on the confocal image was re-imaged with greater resolution with the mitochondrial area outlined (scale bar: 5 μm). The mitochondrial region was straightened as a line scan with the intensity of J2 and 9D5 plotted below.

4°C. Protein concentrations were determined using the Pierce BCA protein assay kit (23225; Thermo Fisher Scientific) before loading on SDS–PAGE gels. Separated proteins were transferred to PVDF membranes (GE10600023; Amersham) and incubated with the specified primary antibodies (Table 1) diluted in PBS containing 5% milk, and 0.5% Tween-20. The membranes were washed and incubated with the appropriate HRP-conjugated secondary antibodies (Dako; Table 1) and visualised for the HRP chemiluminescence using the Amersham ImageQuant 800 Western blot Imaging System (Cytiva) and Bio-Rad ChemiDoc Imaging system.

## MtDNA extraction and copy number quantification

Total DNA containing both mitochondrial and genomic DNA was extracted from cells using the DNeasy Blood and Tissue Kit (69504; QIAGEN) according to the manufacturer's instructions. 2 ng of total DNA per reaction was used to determine the mtDNA copy number. Power SYBR Green PCR Master Mix (4367659; Thermo Fisher Scientific) was used with 500 nM each of forward and reverse primers (Table 2) for DNA amplification using the QuantStudio Real-Time PCR systems (Thermo Fisher Scientific). DNA amplification of mitochondrial sequences was normalised against levels of nuclear *ACTB*; three technical replicates were performed per sample.

## RNA isolation and Northern blotting

For total RNA extraction, TRIzol reagent (15596026; Thermo Fisher Scientific) was added directly to pelleted fibroblast cell lines and RNA was purified from their monolayers according to the manufacturer's instructions. 10 $\mu$g of total cellular RNA was separated on a denaturing 1% formaldehyde agarose gel. RNA was transferred to Hybond-N+ hybridisation membranes (GERPN203B; Sigma-Aldrich) using a vacuum gel transfer system with 10x SSC buffer (1.5 M NaCl, 150 mM sodium citrate) and immobilised by UV-crosslinking. Membranes were hybridised with T7-transcribed [$\alpha$-32P] UTP (Perkin Elmer) radio-labelled riboprobes. Primers used for the transcription of riboprobes are listed in Table 2. Hybridisation was carried out overnight at 65°C in 50% formamide, 7% SDS, 0.2 M NaCl, and 80 mM sodium phosphate (pH 7.4), supplemented with 100 mg/ml salmon sperm DNA. After hybridisation, the membranes were washed for 30 min in a wash buffer containing 0.5x SSC buffer and 0.1% SDS. Imaging was performed using the Typhoon imaging system (GE Healthcare).

## Strand-specific RT-qPCR (ss-RT-qPCR)

The protocol was performed as described previously (Kim et al, 2023). 1 $\mu$g of total RNA per sample was treated with DNase (M6101; Promega). Primers used for strand-specific (SS) reverse transcription

and for RT-qPCR are listed in Table 2. All CMV-tagged strand-specific primers were mixed to a final concentration of 0.1 $\mu$M and used to reverse transcribe mitochondrial and GAPDH transcripts with SuperScript IV Reverse Transcriptase (18090010; Thermo Fisher Scientific). Gene specific forward primers and the CMV-tag reverse primer were used at a final concentration 0.4 $\mu$M with 5 ng of cDNA per reaction for RT-qPCR with SYBR green (A6001; Promega). Three technical replicates were performed per sample.

## Oxygen consumption and extracellular acidification measurements

Rates of oxygen consumption and extracellular acidification were measured using a Seahorse XFe96 Flux Analyzer (Seahorse Biosciences). $4 \times 10^4$ cells were seeded per well of XFe96 cell culture microplates and grown overnight. The next day, cells were incubated with Seahorse assay media (103575-100) supplemented with 10 mM glucose (G8644; Sigma-Aldrich), 2 mM L-Glutamine (25030081; Thermo Fisher Scientific) and 1 mM sodium pyruvate (11360070; Thermo Fisher Scientific) at 37°C for 1 h in a non-$CO_2$ incubator. Inhibitor drug stocks of oligomycin, carbonyl cyanide-p-trifluoromethoxyphenylhydrazone (FCCP), rotenone, and antimycin A from the Mito stress test kit (103015-100) were diluted in supplemented Seahorse assay media. The initial basal oxygen consumption was measured for each well followed by the sequential addition of oligomycin (1.5 $\mu$M), FCCP (1.5 $\mu$M), rotenone (1 $\mu$M), and antimycin A (1 $\mu$M). Each measurement loop consisted of 30 s mixing before measurements were taken every 3 min. After the assay was completed, assay media was removed, and Bradford assay regent (5000006; Bio-Rad) was added directly to the plate to measure protein concentration for normalisation. Four technical replicates were performed per sample.

## Cell proliferation assay

Cell proliferation was measured using an IncuCyte FLR imaging system (Sartorius). $5 \times 10^4$ cells were seeded per well of a 12-well plate and imaged every 4 h for 5–7 d with phase microscopy at 10x magnification. The confluency of each well was quantified by the IncuCyte software. Three technical replicates were performed.

## Colony formation assay

A single cell suspension of $1 \times 10^4$ cells was made and seeded onto a 60 mm cell culture dish, ensuring that the cells were evenly distributed. The cells were incubated for 2 wk. The cells were then washed twice in ice-cold PBS and fixed with ice-cold methanol for 10 min at RT. The fixed monolayer of cells on the dishes was then stained with a crystal violet solution made up of 0.1% crystal violet

---

**(B)** Immunofluorescence of mt-dsRNA with anti-J2 and anti-9D5 antibodies in U2OS cells treated with either IMT1 (10 $\mu$M for 3 h) or RNAse III after fixation. The mitochondrial network was labelled with MitoTracker (MitoT) DeepRed (scale bar: 20 $\mu$m; zoom scale bar: 5 $\mu$m). **(C)** An FFPE section of human colorectal adenocarcinoma was segmented into normal (blue), dysplastic (red), and stroma (yellow, hatched) regions. Subsequent sections of the sample were incubated with anti-9D5 with and without RNase III treatment or anti-ATP5A and visualised with DAB staining (brown). Haematoxylin was used as a counterstain (scale bar 0.5 mm). **(D)** An FFPE section of normal human lung tissue (top row; scale bar 1 mm) and human lung adenocarcinoma (bottom row; scale bar 2 mm) was segmented into normal (blue), adenocarcinoma (red), necrotic (pink), and stroma (yellow, hatched) regions. Subsequent sections of the sample were incubated with anti-9D5 with and without RNase III treatment or anti-ATP5A and visualised with DAB staining (brown). Haematoxylin was used as a counterstain.

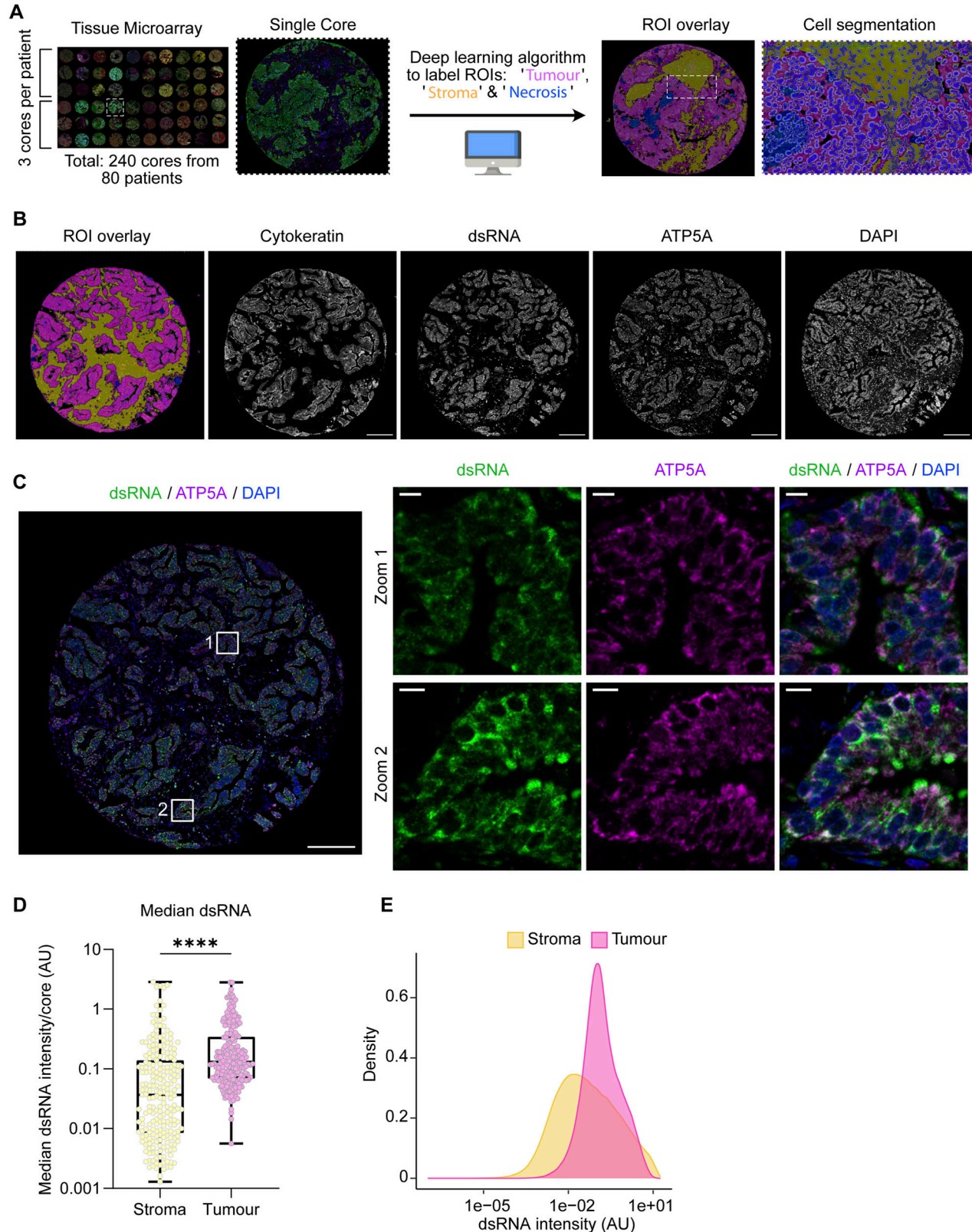

**Figure 6. Mt-dsRNA accumulates in human lung adenocarcinoma.**

**(A)** Lung adenocarcinoma tissue microarrays (TMAs) consisting of three biopsy cores from 80 patients each were used for multiplex staining of cytokeratin, dsRNA, ATP5A, and DAPI. The immunostaining was used to annotate and train a deep learning algorithm to identify regions of interest (ROIs) as "tumour," "stroma," and "necrosis" shown in pink, yellow, and blue overlays, respectively. Individual cells were segmented within the ROIs. **(A, B)** A representative core from lung adenocarcinoma TMA

and 10% ethanol in PBS for 10 min at RT. The cells were then washed three times in PBS, ensuring that only the cell colonies were left stained. The dishes were left to dry overnight at RT before imaging. Two technical replicates were performed per sample.

### In vitro transcription of BrU-labelled RNA and immunodetection

Total DNA extracted from HeLa cells was used as a template to amplify coding and mirror 100 bp regions of *MTCYB*, flanked by a T7 promoter with the Phusion High-Fidelity DNA Polymerase (M0530S; New England Biolabs). The resulting 100 pb PCR products were gel purified and used as a template for T7 RNA transcription. two reactions with T7 polymerase (P2075; Promega) were set up with either a mix of rNTPs at 2.5 mM each or a mix replacing rUTP with BrUTP at a final concentration of 25 mM (18140; Cayman Chemical). These in vitro transcription reactions would produce either ssRNA or ssBrU-RNA, respectively. The corresponding ssRNA pairs were then annealed with the annealing buffer (20 mM Tris pH 7.4, 100 mM NaCl, 0.2 mM $MgCl_2$) with the following programme: 90°C for 10 min followed by consecutive steps of 5°C reduction every 2 min until the temperature reached 4°C. The resulting duplex RNA species were extracted with the addition of TRIzol (15596026; Thermo Fisher Scientific) according to the manufacturer's instructions. Resulting RNA from each species was adjusted to a concentration of 2 μg/μl and 1 μg of the RNA was blotted onto a nitrocellulose membrane and immobilised by UV-crosslinking. Immunodetection was performed with the anti-BrU antibody (Table 1) at a dilution of 1:100.

### Double thymidine block

HeLa cells were seeded on coverslips at a seeding density of $3 \times 10^5$ per well in a six-well culture plate. After 8 h, 2 mM thymidine (T1895; Sigma-Aldrich) was added to the cells and left to incubate for 16 h. The cell monolayer was then washed twice with PBS before the addition of complete DMEM for 9 h. The second block with 2 mM thymidine was then carried out for 16 h. At this point, the cells were blocked at the G1/S boundary. For release from the cell-cycle block, cells were washed twice in PBS and refreshed with complete DMEM for the indicated durations. At the end of each treatment, cells were fixed for immunofluorescence.

### EdU labelling of unsynchronised cells

HeLa cells were seeded on coverslips at a seeding density of $3 \times 10^5$ per well in a six-well plate. After 24 h, cells were fixed using pre-warmed 4% (PFA) in PBS for 15 min, followed by three washes in PBS. The Click-iT EdU Imaging kit (C10086; Life Technologies) was used for the EdU labelling and detection by Alexa Fluor 647 Azide according to the manufacturer's instructions.

### Chromogenic staining of FFPE samples

4 serial sections of the human lung and human colon were cut at 4 μm on TOMO slides and baked for 1 h at 60°C. Staining was performed using the Ventana Discovery Ultra Autostainer (Roche Tissue Diagnostics, RUO Discovery Universal V21.00.0019). Slides were dewaxed onboard, and Discovery CC1 was applied for 32 min at 95°C as antigen retrieval. Each serial section was stained with a different chromogenic indirect immunohistochemical (IHC) assay using the following conditions: anti-dsRNA (9D5) (AB00458-23.0; Absolute antibody) 1/50 was incubated for 32 min, followed by the secondary antibody Discovery Omnimap anti-rabbit HRP (05269679001; Roche Tissue Diagnostics) for 12 min, and detected by the Discovery ChromoMap DAB kit (760-2037; Roche Tissue Diagnostics). For the second assay, the section was pretreated with RNase I (EN0601; Thermo Fisher Scientific) 1/25 for 1 h, anti-dsRNA [9D5] (AB00458-23.0; Absolute antibody) 1/50 was incubated for 32 min, followed by the secondary antibody Discovery Omnimap anti-rabbit HRP (05269679001; Roche Tissue Diagnostics) for 12 min, and detected by the Discovery ChromoMap DAB kit (760-2037; Roche Tissue Diagnostics). For the third assay, sections were pretreated with ShortCut RNase III (M0245S; New England BioLabs) 1/10, anti-dsRNA (9D5) (AB00458-23.0; Absolute antibody) was incubated for 32 min, followed by the secondary antibody Discovery Omnimap anti-rabbit HRP (05269679001; Roche Tissue Diagnostics) for 12 min, and detected by the Discovery ChromoMap DAB kit (760-2037; Roche Tissue Diagnostics). For the fourth assay, recombinant Anti-ATP5A [EPR13030(B)] (ab176569; Abcam) 1/250 was incubated for 32 min, followed by the secondary antibody Discovery Omnimap anti-rabbit HRP (05269679001; Roche Tissue Diagnostics) for 12 min, and detected by the Discovery ChromoMap DAB kit (760-2037; Roche Tissue Diagnostics). Haematoxylin II (05277965001; Roche Tissue Diagnostics) was used as a nuclear counterstain for all four assays. Whole slide images were generated using the Aperio slide scanner (Leica Biosystems) at 40x, and images were imported into Visiopharm (version 2023.01.3.14018) for image analysis. Serial whole slide images were aligned using TissueAlign (version 2023.01.3.14018) and ROI were manually annotated and reviewed by a pathologist.

### Multiplex immunofluorescence staining of FFPE samples

FFPE human lung adenocarcinoma sections were cut at 4 μm on TOMO slides and baked at 60°C for 1 h. Using the Ventana Discovery Ultra autostainer (Roche Tissue Diagnostics, RUO Discovery Universal V21.00.0019). Slides were dewaxed, and antigen retrieval was performed using Discovery CC1 (06414575001; Roche Tissue Diagnostics) at 95°C for 32 min. Antibodies were applied in the following sequence with a denaturing step of CC2 (05279798001; Roche Tissue

showing the ROI overlay as annotated in (A). Individual staining with anti-cytokeratin, anti-dsRNA, anti-ATP5A, and DAPI shown in grayscale (scale bar: 200 μm). **(C)** Representative core from lung adenocarcinoma TMA showing staining with anti-dsRNA (green), anti-ATP5A (magenta), and DAPI (blue) (scale bar: 200 μm). Regions of tumour epithelia are indicated by white boxes which are zoomed in on the right (scale bar: 5 μm). **(D)** Box and whiskers plot of median cytoplasmic dsRNA intensity in stroma and tumour regions per core. Whiskers represent minimum and maximum values. Boxes extend from the 25th to the 75th percentile with the median plotted in the middle (N = 223 cores detected with stroma regions, N = 221 cores detected with tumour regions). The Mann-Whitney t test was used to determine the P-value between stroma versus tumour; P-value < 0.0001 **(E)** Density plot of the distribution of cytoplasmic dsRNA intensity per segmented cell in stroma and tumour ROIs (N = 686,546 cells detected within stroma ROIs, N = 761,319 cells detected within tumour ROIs).

**Table 1.** List of antibodies used and their applications.

| Epitope | Application | Manufacturer |
|---|---|---|
| Primary antibodies | | |
| BrU/BrdU | IF (1:200) dot blot (1:100) | ab6326; Abcam |
| J2-dsRNA | IF (1:200) | 10010500; Scicons |
| TOMM20 | IF (1:300) WB (1:1,000) | ab186734; Abcam |
| SUV3/SUPV3L1 | WB (1:1,000) | sc-365750; Santa Cruz |
| DNA | IF (1:200) | AC-30-10; Progen |
| GRSF1 | IF (1:300) WB (1:1,000) | AV40382; Sigma-Aldrich |
| FASTKD2 | IF (1:300) WB (1:1,000) | 17464-1-AP; ProteinTech |
| PNPase | WB (1:1,000) | sc-365750; Santa Cruz |
| SUV3/SUPV3L1 [C2C3] | IF (1:200) | GTX123034; Genetex |
| TFAM | WB (1:1,000) | 22586-1-AP; ProteinTech |
| NME6 | IF (1:250) WB (1:1,000) | HPA017909; Sigma-Aldrich |
| B-actin–HRP conjugated | WB (1:30,000) | MA5-15739-HRP; Sigma-Aldrich |
| Cyclin A2 | IF (1:200) | ab181591; Abcam |
| mt-HSP70 | WB (1:1,000) | MA3-028; Thermo Fisher Scientific |
| 9D5-dsRNA | IF (1:200) IHC (1:50) | Ab00458-1.1; Absolute Antibody |
| Total human OXPHOS | WB (1:1,000) | ab110411; Abcam |
| ATP5A [EPR13030(B)] | IHC (1:250) | ab176569; Abcam |
| Ki67(30-9) | IHC (1:250) | 790-4286; Roche Tissue Diagnostics |
| Pan-cytokeratin (AE1/AE3) | IHC (1:250) | NCL-L-AE1/AE3-601; Leica Biosystems |
| Secondary antibodies | | |
| Abberior STAR 580 (rat/mouse/rabbit) | IF (1:1,000) | Abberior |
| Abberior STAR RED (rat/mouse/rabbit) | IF (1:1,000) | Abberior |
| Alexa flour-488 IgG (H+L) (rat/mouse/rabbit) | IF (1:1,000) | Thermo Fisher Scientific |
| Alexa flour-594 IgG (H+L) (rat/mouse/rabbit) | IF (1:1,000) | Thermo Fisher Scientific |
| IgG-HRP linked (mouse/rabbit) | WB (1:5,000) | Cell Signalling |
| Discovery Omnimap anti-rabbit HRP | Undiluted | 05269679001; Roche Tissue Diagnostics |
| Opal 570 | IF (1:50) | FP1488001KT; Akoya Biosciences |
| Opal 620 | IF (1:100) | FP1495001KT; Akoya Biosciences |
| Opal 690 | IF (1:300) | FP1497001KT; Akoya Biosciences |
| Opal 780 | IF (1:10) | FP1501001KT; Akoya Biosciences |

Diagnostics) for 24 min between each Opal detection and primary antibody application. Anti-dsRNA (9D5) (AB00458-23.0; Absolute Antibody) was applied at 1/50 and incubated for 32 min at 37°C, followed by the secondary antibody Discovery Omnimap anti-rabbit HRP (05269679001; Roche Tissue Diagnostics) for 12 min, and detected by Opal 570 at 1/50 (FP1488001KT; Akoya Biosciences). Anti-Ki67(30-9) (790-4286; Roche Tissue Diagnostics) was applied for 20 min at 37°C, followed by the secondary antibody Discovery Omnimap anti-rabbit HRP (05269679001; Roche Tissue Diagnostics) for 12 min, and detected by Opal 620 at 1/100 (FP1495001KT; Akoya Biosciences). Recombinant anti-ATP5A (EPR13030(B)) (ab176569; Abcam) was applied at 1/250 and incubated for 32 min, followed by the secondary antibody Discovery Omnimap anti-rabbit HRP (05269679001; Roche Tissue Diagnostics) for 12 min, and detected by

Opal 690 at 1/300 (FP1497001KT; Akoya Biosciences). Anti-pan-cytokeratin (AE1/AE3) (NCL-L-AE1/AE3-601; Leica Biosystems) was applied at 1/250 and was incubated for 28 min, followed by the secondary antibody Discovery Omnimap anti-mouse HRP (05269652001; Roche Tissue Diagnostics) for 12 min. TSA-DIG (FP1502001KT; Akoya Biosciences) was applied at 1/100 for 12 min followed by Opal 780 (FP1501001KT; Akoya Biosciences) at 1/10 for 1 h. QD DAPI (05268826001; Roche Tissue Diagnostics) was applied as a nuclear counterstain.

Each antibody was initially validated in a chromogenic assay and, in single fluorescence, to ensure good specificity and sensitivity. Whole slide images were generated at 20x magnification using MOTIF mode on the PhenoImager HT (Akoya Biosciences). ROI were selected on Phenochart (Akoya Biosciences, version 1.1.0), and

**Table 2. List of primers used.**

| Primer | Forward primer 5′–3′ | Reverse primer 5′–3′ |
|---|---|---|
| Riboprobes (T7 promoter sequence underlined) | | |
| MTND5 | CGGTAATACGACTCACTATAGGGAGA GGCGCAGACTGCTGCGAACA | ACGCCCGAGCAGATGCCAAC |
| MTCYB | CGGTAATACGACTCACTATAGGGAGA GCCTCACGGGAGGACATAGCC | CTCACTCCTTGGCGCCTGCC |
| mirrorCYB | CGGTAATACGACTCACTATAGGGAGA AGACAGTCCCACCCTCACACGA | AATTGTCTGGGTCGCCTAGGAG |
| mirrorND5 | CGGTAATACGACTCACTATAGGGAGA CCCCCATCCTTACCACCCTCGT | GTTGGCATCTGCTCGGGCGT |
| 7SL | CGGTAATACGACTCACTATAGGGAGA AGAGACGGGGTCTCGCTATG | GCCGGGCGCGGTGGCGCGTG |
| mtDNA copy number (gDNA/mtDNA) | | |
| 12 s rRNA | GCACTTAAACACATCTCTGCC | TGAGATTAGTAGTATGGGAGTGG |
| Cytochrome B | CAAACAACCCCCTAGGAATCACC | GTGTTTAAGGGGTTGGCTAGGG |
| Actin | TCACCCACACTGTGCCCATCTACGA | CAGCGGAACCGCTCATTGCCAATGG |
| In vitro transcription of RNA (T7 promoter sequence underlined) | | |
| Cytochrome B | CGGTAATACGACTCACTATAGGGAGA TACTCAGTAGACAGTCCCACC | TGTTTGATCCCGTTTCGTGC |
| Mirror cytochrome B | CGGTAATACGACTCACTATAGGGAGA TGTTTGATCCCGTTTCGTGC | TACTCAGTAGACAGTCCCACC |
| Primers for strand-specific reverse transcription (CMV-tag underlined) | | |
| CMV-GAPDH | CGCAAATGGGCGGTAGGCGTGTGAGCGATGTGGCTCGGCT | |
| CMV-ND4 heavy | CGCAAATGGGCGGTAGGCGTGTGTTTGTCGTAGGCAGATGG | |
| CMV-ND4 light | CGCAAATGGGCGGTAGGCGTGCCTCACACTCATTCTCAACCC | |
| CMV-ND5 heavy | CGCAAATGGGCGGTAGGCGTGTGTTTGGGTTGAGGTGATGATG | |
| CMV-ND5 light | CGCAAATGGGCGGTAGGCGTGCATTGTCGCATCCACCTTTA | |
| CMV-ND6 heavy | CGCAAATGGGCGGTAGGCGTGGGGTTGAGGTCTTGGTGAGTG | |
| CMV-ND6 light | CGCAAATGGGCGGTAGGCGTGCCCATAATCATACAAAGCCCC | |
| CMV-CYTB heavy | CGCAAATGGGCGGTAGGCGTGGGATAGTAATAGGGCAAGGACG | |
| CMV-CYTB light | CGCAAATGGGCGGTAGGCGTGCAATTATACCCTAGCCAACCCC | |
| CMV-CO1 heavy | CGCAAATGGGCGGTAGGCGTGTTGAGGTTGCGGTCTGTTAG | |
| CMV-CO1 light | CGCAAATGGGCGGTAGGCGTGGCCATAACCCAATACCAAACG | |
| CMV-CO2 heavy | CGCAAATGGGCGGTAGGCGTGGGTAAAGGATGCGTAGGGATGG | |
| CMV-CO2 light | CGCAAATGGGCGGTAGGCGTGCTAGTCCTGTATGCCCTTTTCC | |
| 5′–3′ primers for RT-qPCR of strand-specific amplified cDNA | | |
| Reverse CMV-Tag | CGCAAATGGGCGGTAGGCGTG | |
| Forward-GAPDH | CAACGACCACTTTGTCAAGC | |
| Forward-ND4 heavy | CTCACACTCATTCTCAACCCC | |
| Forward-ND4 light | TGTTTGTCGTAGGCAGATGG | |
| Forward-ND5 heavy | CTAGGCCTTCTTACGAGCC | |
| Forward-ND5 light | TAGGGAGAGCTGGGTTGTTT | |
| Forward-ND6 heavy | TCATACTCTTTCACCCACAGC | |
| Forward-ND6 light | TGCTGTGGGTGAAAGAGTATG | |

**Table 2. Continued**

| Primer | Forward primer 5'–3' | Reverse primer 5'–3' |
|---|---|---|
| Forward-CYTB heavy | CAATTATACCCTAGCCAACCCC | |
| Forward-CYTB light | GGATAGTAATAGGGCAAGGACG | |
| Forward-CO1 heavy | GCCATAACCCAATACCAAACG | |
| Forward-CO1 light | TTGAGGTTGCGGTCTGTTAG | |
| Forward-CO2 heavy | CTAGTCCTGTATGCCCTTTTCC | |
| Forward-CO2 light | GTAAAGGATGCGTAGGGATGG | |

spectrally unmixed images were generated using InForm (Akoya Biosciences, version 2.6.0). Unmixed images were quality checked by a pathologist.

All image analysis was carried out on Visiopharm. Tissue microarray core images were de-mapped using the tissue array module. For tissue segmentation, a bespoke deep learning algorithm (version 2024.07.1.16745x64) was trained on annotated images that were verified by a pathologist using cytokeratin, autofluorescence, and DAPI channels. The respective output generated Tumour, Necrosis, Stroma, and Background regions on each core image. Manual corrections were made where necessary. For cell detection, an additional deep learning algorithm (version 2022.12.0.12865) was used that was previously trained using DAPI inputs to generate, background, boundary, and nuclear features. Cytoplasmic labels were generated by dilating nuclear labels by 20 and 10 pixels for tumour and stromal cells, respectively. Output variables were generated each for ROI area, where mean pixel intensities for each marker and X and Y coordinates for each cell were generated and exported for subsequent statistical analysis. Cores with predominant mucinous morphology were excluded, as were any cores, which were predominantly necrosis or lymphoid structures. Antibodies and primers used in this study are detailed in Tables 1 and 2, respectively.

# Supplementary Information

# Acknowledgements

This work was supported by a Cancer Research UK (CRUK) Career Development Fellowship to T MacVicar (RCCFELCDF-May21\100001), Swiss National Science Foundation (31003A_179421) to J-C Martinou, European Union Horizon 2020 Marie Sklodowska-Curie (ITN REMIX 721757) to V Xavier, the Novartis Foundation for Medical-Biological Research to V Xavier, Mazumdar-Shaw Chair endowment funding to J Le Quesne, CRUK Core funding to LM Carlin (CRUK A23983) and CRUK core funding to the CRUK Scotland Institute (A17196 and A31287). The CRUK Scotland Institute acknowledges the support of NHS Research Scotland (NRS) Greater Glasgow and Clyde Biorepository. We thank Ashish Dhir and Alexis Barr (Imperial College London) for insightful discussion. We also thank the Beatson Advanced Imaging Resource and the Bioimaging Core Facility at University of Geneva. The manuscript was critically reviewed by Catherine Winchester.

## Author Contributions

V Xavier: conceptualization, formal analysis, validation, investigation, visualization, methodology, and writing—original draft, review, and editing.
S Martinelli: data curation, validation, investigation, and methodology.
R Corbyn: software, visualization, and methodology.
R Pennie: investigation and methodology.
K Rakovic: data curation, validation, and writing—review and editing.
IR Powley: data curation, formal analysis, validation, and writing—review and editing.
L Officer-Jones: supervision, validation, methodology, and writing—review and editing.
V Ruscica: investigation and writing—review and editing.
A Galloway: investigation and writing—review and editing.
LM Carlin: resources, supervision, and writing—review and editing.
VH Cowling: supervision and validation.
J Le Quesne: resources, supervision, funding acquisition, validation, and methodology.
J-C Martinou: conceptualization, resources, supervision, funding acquisition, validation, project administration, and writing—review and editing.
T MacVicar: conceptualization, formal analysis, supervision, funding acquisition, validation, investigation, visualization, project administration, and writing—original draft, review, and editing.

## Conflict of Interest Statement

The authors declare that they have no conflict of interest.

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
