## [Reviewer comments · Life Science Alliance]

Life Science Alliance

Mitochondrial double-stranded RNA homeostasis depends on cell-cycle progression

Vanessa Xavier, Silvia Martinelli, Ryan Corbyn, Rachel Pennie, Kai Rakovic, Ian Powley, Leah Officer-Jones, Vincenzo Ruscica, Alison Galloway, Leo Carlin, Victoria Cowling, John Le Quesne, Jean-Claude Martinou, and Thomas MacVicar

DOI: [10.26508/lsa.202402764](https://doi.org/10.26508/lsa.202402764)

Corresponding author(s): Thomas MacVicar, CRUK Scotland Institute and Jean-Claude Martinou, University of Geneva

Review Timeline:

Submission Date:	2024-04-10
Editorial Decision:	2024-05-06
Revision Received:	2024-08-15
Editorial Decision:	2024-08-16
Revision Received:	2024-08-19
Accepted:	2024-08-21

Transaction Report:

May 6, 2024

Re: Life Science Alliance manuscript #LSA-2024-02764-T

Dr. Thomas MacVicar
CRUK Scotland Institute
Switchback Road
Glasgow G61 1BD
United Kingdom

Dear Dr. MacVicar,

Thank you for submitting your manuscript entitled "Mitochondrial double-stranded RNA homeostasis depends on cell-cycle progression" to Life Science Alliance. The manuscript was assessed by expert reviewers, whose comments are appended to this letter. We invite you to submit a revised manuscript addressing the Reviewer comments.

Thank you for this interesting contribution to Life Science Alliance. We are looking forward to receiving your revised manuscript.

Sincerely,

B. MANUSCRIPT ORGANIZATION AND FORMATTING:

Reviewer #1 (Comments to the Authors (Required)):

The manuscript by Xavier et al investigated the localization and homeostasis of mt-dsRNAs. The authors showed that mt-dsRNAs are largely associated with the RNA progressing granules within the mitochondria and their abundance depends on the supply of mitochondrial ribonucleotide. In addition, they characterized the expression of mt-dsRNAs and showed increased RNA levels in cancer as their expression is associated with cell proliferation.

For the most part, the manuscript presents interesting observations on mt-dsRNA localization inside mitochondria and reports a potential link between mt-dsRNA expression and proliferation. This reviewer's specific comments on the manuscript are the following:

Major comments

- 1) Previously, it was suggested that light-strand mtRNAs were degraded quickly after transcription, which prevents the accumulation of mt-dsRNAs. Using STED microscopy, can authors analyze the colocalization between BrU-RNA and SUV3?
- 2) Similarly, what would happen with the degree of colocalization between dsRNA foci and BrU RNA when SUV3 is depleted?
- 3) Authors showed increased mt-dsRNAs in transformed cells. Considering that mt-dsRNAs can induce an inflammatory response, is there any link between mt-dsRNA levels and the basal levels of interferon / interferon-stimulated genes?
- 4) The authors claim that increased expression of NME6 is responsible for increased mt-dsRNAs in transformed cells. What happens to mt-dsRNA levels when NME6 is overexpressed or rNTP is supplied in WT fibroblasts?
- 5) What happens to the expression of mitochondrial RNA polymerase (POLRMT) when cell proliferation is stimulated? Do authors have a model that can account for increased transcription of mtRNAs when cell proliferation is stimulated?

Minor comment

- 1) In the introduction, please cite Yoon et al, *Mol Ther-Nucleic Acids* (2022) that showed the role of mt-dsRNAs in autoimmune disease

Reviewer #2 (Comments to the Authors (Required)):

Mitochondria are a unique organelle that contains its own circular DNA genome. Transcription occurs in both directions, resulting in mitochondrial dsRNA. This is a descriptive manuscript that is largely based on advanced imaging, describing (1) the sub-organellar localisation of mt-dsRNA and (2) situations in which mt-dsRNA levels are increased or decreased, including transformation, cell cycle progression, and altered nucleotide levels. Given that mitochondria play essential roles in metabolism and immunity (including via release of mt-dsRNA), and are important in many human diseases, these observations are of high importance to several fields. The manuscript is very well written and contains nicely laid out figures. Overall, the conclusions are well supported by the data. However, this reviewer has no expertise in imaging. Taken together, this manuscript is a strong candidate for LSA if the following points can be addressed.

Major points.

1. The authors need to validate the specificity of the J2 antibody signal using RNase III, as was done in Fig 5 for the 9D5 antibody. Can the authors test how changes in the signal obtained from J2 staining correlate with dsRNA levels?
2. With the notable exception of Fig 2C, mt-dsRNA detection is based on antibody detection. The manuscript should be strengthened by including northern blot analysis in Fig 4, and/or by performing strand-specific RT-qPCR or RNAseq for key experiments.
3. Fig 5C/D. The number of analysed tumour samples is very small. The authors should analyse samples from multiple patients to demonstrate that their conclusion is generalisable. If this is not possible, a very clear statement that this is a limitation of the study needs to be provided.

Minor points

1. Antibodies used in immunofluorescence detection of proteins should be validated using siRNA knockdowns. This is already included for SUV3 and NME6 but is missing for FASTKD2, GRSF1, and cyclin A.
1. Figs 2D and 5 should include quantifications.
2. Fig 3C. Is the colour scheme of the label wrong? dsRNA appears to be in white, not red.
3. Page 4. The authors claim that "Mt-dsRNA accumulates within foci in mitochondria lacking functional mtRNA degradosomes (Dhir et al., 2018)" However, Dhir et al observe mtRNA foci in HeLa cells that have a functional degradosome (Fig 1c), and the light microscopy techniques used in the paper do not allow the visualisation of foci inside mitochondria. The EM data show a J2 signal inside mitochondria but no clear foci. Thus, the authors should cite Dhir et al as follows: "mt-dsRNA foci were detected in cultured cells with normal mtRNA processing, and are enhanced in cells lacking functional mtRNA degradosomes".
4. Fig S1A. There is a typo (dsRNA instead of ssRNA, condition 2).

Reviewer #3 (Comments to the Authors (Required)):

In this manuscript the authors suggest that the levels of double-stranded (ds) mitochondrial RNA is linked to the cell cycle. Previous work show that ds mitochondrial RNA accumulate in response to various defects of mitochondrial RNA processing. Here the authors demonstrate that this accumulation is also linked to the cell cycle.

This is a well written manuscript, using the combination of high-resolution and confocal microscopy and cell lines as well as human tumour tissue.

The presented data seem to suggest that the accumulation of ds mitochondrial RNA is a consequence of increased de novo transcription in mitochondria. Such a conclusion could be supported by measuring de novo transcription rates.

As a minor point, in figure 5D, it would be beneficial if the authors could highlight what difference the reader should look at, it is not very clear at the moment.

Reviewer #1 (Comments to the Authors (Required)):

The manuscript by Xavier et al investigated the localization and homeostasis of mt-dsRNAs. The authors showed that mt-dsRNAs are largely associated with the RNA progressing granules within the mitochondria and their abundance depends on the supply of mitochondrial ribonucleotide. In addition, they characterized the expression of mt-dsRNAs and showed increased RNA levels in cancer as their expression is associated with cell proliferation.

For the most part, the manuscript presents interesting observations on mt-dsRNA localization inside mitochondria and reports a potential link between mt-dsRNA expression and proliferation. This reviewer's specific comments on the manuscript are the following:

We thank the reviewer for their supportive critique of our work.

Major comments

1) Previously, it was suggested that light-strand mtRNAs were degraded quickly after transcription, which prevents the accumulation of mt-dsRNAs. Using STED microscopy, can authors analyze the colocalization between BrU-RNA and SUV3?

As the reviewer points out, light strand mtRNA appears to be degraded rapidly by the degradosome (Borowski *et al*, 2012; Dhir *et al*, 2018; Szczesny *et al*, 2010). Co-localisation was previously observed between nascent mtRNA (BrU labelled) and exogenously expressed SUV3-PNPase complex (Borowski *et al.*, 2012). We analysed the degree of overlap between BrU (ssRNA) and endogenous SUV3 foci (Fig. 1F; Fig. S1D) and describe the result in the text: "*SUV3 foci were variable in size and the majority of small SUV3 foci did not colocalise with ssRNA, dsRNA or GRSF1 (Fig.S1E). However, large SUV3 foci overlapped with dsRNA, ssRNA and GRSF1 to a similar degree and approximately 50% of dsRNA foci overlapped partially with SUV3 (Fig. 1F; Fig. S1D).*"

2) Similarly, what would happen with the degree of colocalization between dsRNA foci and BrU RNA when SUV3 is depleted?

We depleted SUV3 by siRNA and immunolabelled BrU-RNA and dsRNA in U2OS cells prior to STED imaging (Rev. Fig. 1A). Interestingly, knockdown of SUV3 resulted in enlarged dsRNA foci (Rev. Fig. 1B) but smaller BrU foci (Rev. Fig. 1C). The reduced intensity of BrU-labelled de novo transcripts indicates that transcription is suppressed in cells lacking SUV3, which is consistent with recently published imaging and *in organello* transcription data (Zhu *et al*, 2022). These data are in line with a role for SUV3 in maintaining mitochondrial ssRNA and dsRNA homeostasis. Consequently, an increased size of dsRNA foci upon SUV3 downregulation resulted in greater overlap between dsRNA and BrU. (Rev. Fig. 1A, D). We decided not to incorporate these data into the manuscript, which remains focussed on the accumulation of mitochondrial dsRNA in proliferating cells with unperturbed SUV3 and PNPase.

Rev. Fig. 1

Rev. Fig.1

(A) U2OS cells were treated with 20nM of the indicated siRNA for 72hours and pulsed with BrU for 1 h. Cells were then immunostained with anti-BrU (green), anti-dsRNA (red) and anti-TOMM20 antibodies (grey) and imaged with STED microscopy (scale bar: 2 μm). (B) Scatter plot of dsRNA foci area from representative images in (A). (number of foci measured: siLuc = 105, siSUV3 = 154; number of cells = 2 from one culture). The Mann-Whitney test was used to determine the P-value between siLuc vs siSUV3 P-Value <0.0001. Horizontal lines indicate the mean value and error bars indicate the standard deviation. (C) Scatter plot of BrU foci area from representative images in (A). (number of foci measured: siLuc = 105, siSUV3 = 173; number of cells = 2 from one culture). The Mann-Whitney test was used to determine the P-value between siLuc vs siSUV3 P-Value <0.0001. Horizontal lines indicate the mean value and error bars indicate the standard deviation. (D) The percentage of BrU overlapped by dsRNA foci categorised as in (Manuscript Fig. 1C; number of foci measured: siLuc = 105, siSUV3 = 154; number of cells = 2 from one culture).

3) Authors showed increased mt-dsRNAs in transformed cells. Considering that mt-dsRNAs can induce an inflammatory response, is there any link between mt-dsRNA levels and the basal levels of interferon / interferon-stimulated genes?

As the reviewer points out, several studies have shown that mt-dsRNA can trigger an inflammatory response once exposed to the cytosol. It is indeed interesting to consider whether the accumulation of dsRNA within mitochondria of transformed cells is sufficient to alter inflammatory signalling pathways. As suggested by the reviewer, we examined interferon beta (IFN β) expression in WT fibroblasts (low dsRNA), hTERT fibroblasts (low dsRNA) and hTERT LT fibroblasts (high dsRNA). Basal expression of IFN β was elevated in hTERT LT fibroblasts (Rev. Fig. 2A). While this response correlated with enhanced mt-dsRNA levels, it was not sensitive to mtDNA depletion caused by treatment with dideoxycytidine (ddC) (Rev. Fig. 2B). We therefore conclude that any increase in basal inflammation signalling in these transformed cells is independent of mitochondrial nucleic acid, including mt-dsRNA, and instead may stem from SV40-LT expression (Forero *et al*, 2014).

We also tested the impact of mt-dsRNA on basal inflammation in an endothelial cell line (SVEC-10) that is commonly used to monitor nucleic acid-induced innate immune responses. We used CRISPR to deplete NME6 in these cells, which reduced the levels of mt-dsRNA (Rev. Fig. 2C) but did not see any effect on IFN β or interferon stimulated gene 15 (ISG15) expression (Rev. Fig. 2D).

Therefore, accumulation of mt-dsRNA is insufficient to induce an inflammatory response in the cell lines that we tested in the absence of stress/stimuli that trigger the cytosolic release of mt-dsRNA.

Rev. Fig. 2

Rev. Fig.2. (A) IFN- β transcript levels analysed by qRT-PCR in WT, hTERT and hTERT LT fibroblasts. (n = 2 independent cultures). (B) IFN- β transcript levels analysed by qRT-PCR in hTERT LT fibroblasts that were treated with DMSO or 200 μ M of 2'-3'-dideoxycytidine (ddC) for 48 hours (n = 2 independent cultures). (C) SVEC WT and NME6 KO cells were immunostained for dsRNA (green) and TOMM20 (magenta). DAPI staining shown in blue. (Scale bar: 20 μ m). (D) IFN- β and ISG15 transcript levels analysed by qRT-PCR in SVEC WT (WT) and NME6 KO (KO) cells (n = 3 independent cultures).

4) The authors claim that increased expression of NME6 is responsible for increased mt-dsRNAs in transformed cells. What happens to mt-dsRNA levels when NME6 is overexpressed or rNTP is supplied in WT fibroblasts?

While NME6 levels are enhanced in transformed cells (Fig. S2E), we refrain from stating that increased NME6 expression is responsible for enhanced mt-dsRNA per se. Instead, more NME6 is likely required for the supply of NTPs to meet an increased demand on mitochondrial transcription during/following mitochondrial biogenesis. As suggested by the reviewer, we treated WT fibroblasts with exogenous nucleosides and did not observe an increase in mt-dsRNA (New Fig. S3D). We have also adjusted the text to report these data: *“However, nucleoside supplementation in WT fibroblasts did not increase mt-dsRNA levels, which indicates that boosting mitochondrial rNTPs alone is insufficient to drive mt-dsRNA foci formation (Fig. S3C). Collectively, these data reveal that mt-dsRNA accumulates in rapidly dividing malignant cells, which is supported by the supply of mitochondrial rNTPs by NME6 for RNA synthesis.”*

5) What happens to the expression of mitochondrial RNA polymerase (POLRMT) when cell proliferation is stimulated? Do authors have a model that can account for increased transcription of mtRNAs when cell proliferation is stimulated?

We treated fibroblasts with FGF or FGF + aphidicolin and determined the level of POLRMT and other mitochondrial proteins by immunoblot (New Fig. 4G). We now state in the manuscript: *“Levels of the mitochondrial RNA polymerase (POLRMT), NME6, PNPase and SUV3 were unchanged upon FGF treatment in hTERT fibroblasts (Fig. 4G), which indicates that acute upregulation of mitochondrial RNA synthesis and mt-dsRNA is not a consequence of altered expression of mitochondrial transcription machinery or degradosome components”*.

We do not yet understand how mitochondrial transcription is regulated following the acute induction of cell proliferation but this is indeed an exciting avenue for future work (please see also response to Reviewer 3). A broader analysis of the mitochondrial proteome may reveal changes in other regulators of transcription or mtRNA homeostasis in FGF-treated cells. Interestingly, recent work in *Drosophila* has highlighted control of mitochondrial gene expression and biogenesis in response to EGF signalling (Zhang *et al*, 2022).

Minor comment

1) In the introduction, please cite Yoon *et al*, *Mol Ther-Nucleic Acids* (2022) that showed the role of mt-dsRNAs in autoimmune disease

We now cite this work: *“Consequently, mt-dsRNA is implicated in the induction of inflammatory responses in diverse pathophysiological scenarios such as osteoarthritis (Kim *et al*, 2022), alcohol liver disease (Lee *et al*, 2020), chronic kidney disease (Zhu *et al*, 2023) and autoimmune diseases (Hooftman *et al*, 2023; Yoon *et al*, 2022).”*

Reviewer #2 (Comments to the Authors (Required)):

Mitochondria are a unique organelle that contains its own circular DNA genome. Transcription occurs in both directions, resulting in mitochondrial dsRNA. This is a descriptive manuscript that is largely based on advanced imaging, describing (1) the sub-organellar localisation of mt-dsRNA and (2) situations in which mt-dsRNA levels are increased or decreased, including transformation, cell cycle progression, and altered nucleotide levels. Given that mitochondria play essential roles in metabolism and immunity (including via release of mt-dsRNA), and are important in many human diseases, these observations are of high importance to several fields. The manuscript is very well written and contains nicely laid out figures. Overall, the conclusions are well supported by the data. However, this reviewer has no expertise in imaging. Taken together, this manuscript is a strong candidate for LSA if the following points can be addressed.

We thank the reviewer for their positive assessment of our work.

Major points.

1. The authors need to validate the specificity of the J2 antibody signal using RNase III, as was done in Fig 5 for the 9D5 antibody. Can the authors test how changes in the signal obtained from J2 staining correlate with dsRNA levels?

We now show that J2 immunofluorescence signal is sensitive to RNase III and the inhibitor of mitochondrial transcription, IMT1 (New Fig. 5B, New Fig. S5). Importantly, J2-labelled mitochondrial dsRNA foci are absent in samples treated with RNase III and IMT1. The specificity of the J2 antibody for dsRNA and its interaction with mt-dsRNA has been tested extensively in previous publications by other approaches including J2-immunoprecipitation-based dsRNA sequencing (Dhir *et al.*, 2018). While we cannot determine the concentration of dsRNA using the antibody alone, our antibody validation tests and experiments (including the accumulation of dsRNA in cells lacking SUV3 and PNPase, which is now quantified in New Fig. S3B), show that the J2 immunofluorescence signal responds as expected to alterations in mt-dsRNA levels. We are therefore confident in the performance of J2, and 9D5 to report the localisation and relative abundance of mt-dsRNA in cells and tissue.

2. With the notable exception of Fig 2C, mt-dsRNA detection is based on antibody detection. The manuscript should be strengthened by including northern blot analysis in Fig 4, and/or by performing strand-specific RT-qPCR or RNAseq for key experiments.

We agree with the reviewer that alternative methods of dsRNA detection can complement our anti-dsRNA immunofluorescence experiments. We therefore performed strand-specific RT-qPCR to determine levels of multiple heavy and light mtRNA transcripts in a semi-quantitative manner (Kim *et al.*, 2023). Indeed, we found that FGF treatment of fibroblasts increased both heavy and light transcripts, which was reversed by co-treatment with aphidicolin (New. Fig.4E). This was also backed up by

Northern blot (New. Fig. S4B). These data are consistent with our immunofluorescence analysis (Fig. 4B,D) and strengthen the major conclusion from our work that mt-dsRNA is enhanced in proliferating cells. We have edited the text as follows: “*Consistent with the immunodetection of dsRNA, strand-specific RT-qPCR (Kim et al., 2023) revealed an accumulation of heavy and light strand transcripts in FGF-treated fibroblasts, which was reversed by co-treatment with aphidicolin (Fig. 4E). Similarly, Northern blot analysis of the coding and mirror transcripts of CYTB revealed increased transcript levels from both strands of mtDNA in FGF-treated fibroblasts (Fig. S4B).*”

3. Fig 5C/D. The number of analysed tumour samples is very small. The authors should analyse samples from multiple patients to demonstrate that their conclusion is generalisable. If this is not possible, a very clear statement that this is a limitation of the study needs to be provided.

We expanded our analysis greatly of dsRNA in tumour tissue by multiplex staining 3 cores of tissue per lung adenocarcinoma patient in a tumour microarray that consists of tissue from 80 patients (New. Fig.6). We used supervised deep learning to quantify the intensity of dsRNA in tumour and stromal cells. We could confirm significant overlap between the mitochondrial marker ATP5A and dsRNA in the new tumour samples. These data demonstrate that, on average, tumour regions contain more dsRNA than stromal regions. We report these new data in the text on page 10-11.

Minor points

1. Antibodies used in immunofluorescence detection of proteins should be validated using siRNA knockdowns. This is already included for SUV3 and NME6 but is missing for FASTKD2, GRSF1, and cyclin A.

The anti-FASTKD2 and anti-GRSF1 antibodies are used routinely as bone fide markers of MRGs by immunofluorescence and have been validated by us and others (Antonicka & Shoubridge, 2015; Jourdain *et al*, 2013). Anti-Cyclin-A2 is used routinely in cell cycle research (56 publications).

2. Figs 2D and 5 should include quantifications.

We now include quantification for Fig.2D (New. Fig.S3B) and for the updated Fig.5B (New. Fig.S5). Our multiplex analysis of tumour tissue (formerly Fig. 5E) is quantified in New. Fig. 6.

3. Fig 3C. Is the colour scheme of the label wrong? dsRNA appears to be in white, not red.

Thank you for pointing out this error. We have edited the labelling.

4. Page 4. The authors claim that "Mt-dsRNA accumulates within foci in mitochondria lacking functional mtRNA degradosomes (Dhir et al., 2018)" However, Dhir et al observe mtRNA foci in HeLa cells that have a functional degradosome (Fig 1c), and the light

microscopy techniques used in the paper do not allow the visualisation of foci inside mitochondria. The EM data show a J2 signal inside mitochondria but no clear foci. Thus, the authors should cite Dhir et al as follows: "mt-dsRNA foci were detected in cultured cells with normal mtRNA processing, and are enhanced in cells lacking functional mtRNA degradosomes".

We agree and have edited the text according to the reviewer's recommendation: "*Mt-dsRNA foci were detected in cultured cells with normal mtRNA processing, and are enhanced in cells that lack functional mtRNA degradosomes (Dhir et al., 2018). Mt-dsRNA also accumulates in the absence of other regulators of mtRNA processing, including the exoribonuclease REXO2 (Szewczyk et al, 2020) and the RNA binding protein GRSF1 (Hensen et al, 2019).*"

5. Fig S1A. There is a typo (dsRNA instead of ssRNA, condition 2).

Thank you for pointing out this error. We have edited the labelling.

Reviewer #3 (Comments to the Authors (Required)):

In this manuscript the authors suggest that the levels of double-stranded (ds) mitochondrial RNA is linked to the cell cycle. Previous work show that ds mitochondrial RNA accumulate in response to various defects of mitochondrial RNA processing. Here the authors demonstrate that this accumulation is also linked to the cell cycle. This is a well written manuscript, using the combination of high-resolution and confocal microscopy and cell lines as well as human tumour tissue.

We thank the reviewer for their positive assessment of our work.

The presented data seem to suggest that the accumulation of ds mitochondrial RNA is a consequence of increased de novo transcription in mitochondria. Such a conclusion could be supported by measuring de novo transcription rates.

We agree with the reviewer that our BrU labelling data suggest an accumulation of mt-dsRNA is the result of increased de novo mitochondrial transcription in proliferating cells. At this stage we do not have a model for how transcription may be increased upon stimulation of cell proliferation (see Reviewer 1 point 5). Our pulse BrU labelling for 1 h is not a measurement of transcription rate per se and could instead reflect enhanced stability of nascent transcripts or a greater proportion of nucleoids engaged in transcription. We include some minor speculation in the discussion (page 12) and cite previous work that argues transcription rates are highest at G1 and G2 of cell cycle. Nevertheless, we refrain from suggesting that transcription rate is increased here and instead believe this could be studied in more depth in future work.

As a minor point, in figure 5D, it would be beneficial if the authors could highlight what difference the reader should look at, it is not very clear at the moment.

We agree and have edited Fig. 5C, D and the respective legends for clarity.

Rebuttal References

- Antonicka H, Shoubridge Eric A (2015) Mitochondrial RNA Granules Are Centers for Posttranscriptional RNA Processing and Ribosome Biogenesis. *Cell Reports* 10: 920-932
- Borowski LS, Dziembowski A, Hejnowicz MS, Stepień PP, Szczesny RJ (2012) Human mitochondrial RNA decay mediated by PNPase-hSuv3 complex takes place in distinct foci. *Nucleic Acids Research* 41: 1223-1240
- Dhir A, Dhir S, Borowski LS, Jimenez L, Teitell M, Rötig A, Crow YJ, Rice GI, Duffy D, Tamby C *et al* (2018) Mitochondrial double-stranded RNA triggers antiviral signalling in humans. *Nature* 560: 238-242
- Forero A, Giacobbi NS, McCormick KD, Gjoerup OV, Bakkenist CJ, Pipas JM, Sarkar SN (2014) Simian Virus 40 Large T Antigen Induces IFN-Stimulated Genes through ATR Kinase. *The Journal of Immunology* 192: 5933-5942
- Hensen F, Potter A, van Esveld SL, Tarrés-Solé A, Chakraborty A, Solà M, Spelbrink JN (2019) Mitochondrial RNA granules are critically dependent on mtDNA replication factors Twinkle and mtSSB. *Nucleic Acids Res* 47: 3680-3698
- Hooftman A, Peace CG, Ryan DG, Day EA, Yang M, McGettrick AF, Yin M, Montano EN, Huo L, Toller-Kawahisa JE *et al* (2023) Macrophage fumarate hydratase restrains mtRNA-mediated interferon production. *Nature*
- Jourdain AA, Koppen M, Wydro M, Rodley CD, Lightowlers RN, Chrzanowska-Lightowlers ZM, Martinou JC (2013) GRSF1 regulates RNA processing in mitochondrial RNA granules. *Cell Metab* 17: 399-410
- Kim S, Lee K, Choi YS, Ku J, Kim H, Kharbush R, Yoon J, Lee YS, Kim J-H, Lee YJ *et al* (2022) Mitochondrial double-stranded RNAs govern the stress response in chondrocytes to promote osteoarthritis development. *Cell Reports* 40: 111178
- Kim S, Yoon J, Lee K, Kim Y (2023) Analysis of mitochondrial double-stranded RNAs in human cells. *STAR Protoc* 4: 102007
- Lee J-H, Shim Y-R, Seo W, Kim M-H, Choi W-M, Kim H-H, Kim YE, Yang K, Ryu T, Jeong J-M *et al* (2020) Mitochondrial Double-Stranded RNA in Exosome Promotes Interleukin-17 Production Through Toll-Like Receptor 3 in Alcohol-associated Liver Injury. *Hepatology* 72: 609-625
- Szczesny RJ, Borowski LS, Brzezniak LK, Dmochowska A, Gewartowski K, Bartnik E, Stepień PP (2010) Human mitochondrial RNA turnover caught in flagranti: involvement of hSuv3p helicase in RNA surveillance. *Nucleic Acids Res* 38: 279-298
- Szewczyk M, Malik D, Borowski LS, Czarnomska Sylwia D, Kotrys AV, Klosowska-Kosicka K, Nowotny M, Szczesny RJ (2020) Human REXO2 controls short mitochondrial RNAs generated by mtRNA processing and decay machinery to prevent accumulation of double-stranded RNA. *Nucleic Acids Research* 48: 5572-5590
- Yoon J, Lee M, Ali AA, Oh YR, Choi YS, Kim S, Lee N, Jang SG, Park S, Chung JH *et al* (2022) Mitochondrial double-stranded RNAs as a pivotal mediator in the pathogenesis of Sjögren's syndrome. *Mol Ther Nucleic Acids* 30: 257-269
- Zhang C, Jin Y, Marchetti M, Lewis MR, Hammouda OT, Edgar BA (2022) EGFR signaling activates intestinal stem cells by promoting mitochondrial biogenesis and β -oxidation. *Current Biology* 32: 3704-3719.e3707

Zhu X, Xie X, Das H, Tan BG, Shi Y, Al-Behadili A, Peter B, Motori E, Valenzuela S, Posse V *et al* (2022) Non-coding 7S RNA inhibits transcription via mitochondrial RNA polymerase dimerization. *Cell*

Zhu Y, Zhang M, Wang W, Qu S, Liu M, Rong W, Yang W, Liang H, Zeng C, Zhu X *et al* (2023) Polynucleotide phosphorylase protects against renal tubular injury via blocking mt-dsRNA-PKR-eIF2 α axis. *Nature Communications* 14: 1223

August 16, 2024

RE: Life Science Alliance Manuscript #LSA-2024-02764-TR

Thomas MacVicar

Dear Dr. MacVicar,

Thank you for submitting your revised manuscript entitled "Mitochondrial double-stranded RNA homeostasis depends on cell-cycle progression". We would be happy to publish your paper in Life Science Alliance pending final revisions necessary to meet our formatting guidelines.

- please be sure that the authorship listing and order is correct
- please add ORCID ID for secondary corresponding author-instructions on how to do so have been sent to them
- please add keywords and a category for your manuscript to our system
- please add the author contributions to the main manuscript text; -please consult our manuscript preparation guidelines <https://www.life-science-alliance.org/manuscript-prep> for the correct order of your manuscript

Figure Check:

- please add sizes next to blots in Figure 2C and S1C

LSA now encourages authors to provide a 30-60 second video where the study is briefly explained. We will use these videos on social media to promote the published paper and the presenting author (for examples, see <https://docs.google.com/document/d/1-UWCfbE4pGcDdcgzcmiuJl2XMBJnxKYeqRvLLrLSo8s/edit?usp=sharing>). Corresponding or first-authors are welcome to submit the video. Please submit only one video per manuscript. The video can be emailed to contact@life-science-alliance.org

A. FINAL FILES:

B. MANUSCRIPT ORGANIZATION AND FORMATTING:

Sincerely,

August 21, 2024

RE: Life Science Alliance Manuscript #LSA-2024-02764-TRR

Dr. Thomas MacVicar
CRUK Scotland Institute
Garscube Estate
Switchback Road
Glasgow G61 1BD
United Kingdom

Dear Dr. MacVicar,

Thank you for submitting your Research Article entitled "Mitochondrial double-stranded RNA homeostasis depends on cell-cycle progression". It is a pleasure to let you know that your manuscript is now accepted for publication in Life Science Alliance. Congratulations on this interesting work.

DISTRIBUTION OF MATERIALS:

Again, congratulations on a very nice paper. I hope you found the review process to be constructive and are pleased with how the manuscript was handled editorially. We look forward to future exciting submissions from your lab.

Sincerely,
